- Effects of Seasonal Snow Cover on Hydrothermal Conditions of the Active Layer in
- the Northeastern Qinghai-Tibet Plateau
- Ji Chen<sup>1,2</sup>, Yu Sheng<sup>1</sup>, Qingbai Wu<sup>1,2</sup>, Lin Zhao<sup>3</sup>, Jing Li<sup>1</sup>, Jingyi Zhao<sup>1,2</sup>
- <sup>1</sup>State Key Laboratory of Frozen Soil Engineering, Cold and Arid Regions Environmental and
- Engineering Research Institute, Chinese Academy of Science, Lanzhou, 730000, China
- <sup>2</sup> Beiluhe Observation Station of Frozen soil Environment and Engineering, Cold and Arid
- Regions Environmental and Engineering Research Institute, Chinese Academy of Science,
- Lanzhou, 730000, China
- <sup>3</sup>Cryosphere Research Station on the Qinghai-Tibetan Plateau, Chinese Academy of Sciences,
- Lanzhou, 730000, China
- Corresponding to: chenji@lzb.ac.cn
- Abstract. Snow cover significantly influences the moisture and thermal properties of the active
- layer in permafrost regions. Seasonal snow cover, soil temperature, and moisture were monitored
- in the northeastern Qinghai-Tibet Plateau (QTP) from December 2012 to February 2015.
- According to field data, the following conclusions were drawn. (1) The snow season in this region
- is predominantly during spring (March to May) and autumn (September to November), the
- thickness of individual snowfall events is usually less than 5 cm, and the duration of land surface
- snow cover is generally no longer than 5 days. (2) Removal of seasonal snow cover is beneficial
- for cooling the active layer in a whole year and in other seasons with the exception of summer.
- Further analysis on the ground temperature in the active layer shows that the cooling effect of the
- snow removal maybe results from the high thermal resistivity of snow, the delay of snowfall time
- in autumn, and the drastic decrease of moisture content in the active layer. (3) Seasonal snow
- cover maintains the high water content of the active layer. Snow removal can therefore lead to a
- rapid decrease of soil moisture content. A small decrease in water content of the active layer at the
- natural snow site (NSS) is related with less rainfall during the monitoring period. Significant
- differences between the NSS and the snow removal site (SRS) may depend predominantly on the
- inhibitory action of snow cover on the evaporation capacity of surface soil because of its cooling
- and shading effects during the daytime and in summer.
- Keywords: Seasonal snow cover, Active layer, Soil temperature, Soil moisture, Qinghai-Tibet
- Plateau

### 31 1. Introduction

The active layer is defined as the top layer of ground that is subject to annual thawing and 33 freezing in areas underlain by permafrost (Washburn, 1979). The active layer over permafrost 34 plays a significant role in the surface energy balance, the hydrologic cycle, carbon exchange 35 between the atmosphere and the land surface, ecosystems, landscape processes, and human 36 infrastructure in cold regions (Brownet al., 2000; Lemkeet al., 2007; Wang et al., 2009; Han et al., 37 2010). Due to the impact of global climate change and human engineering activities, active layer 38 thickness and temperature have increased over the past few decades in the Arctic, Antarctic, 39 Alpine, QTP, and other areas (Brown et al., 2000; Jin et al., 2000; Zhao et al., 2000; Harris et al., 40 2003; Nelson et al., 2004; Zhao et al., 2004; Wu and Liu, 2004; Zhang et al., 2005; Cheng and Wu, 41 2007; Zhao et al., 2008; Wu and Zhang, 2010; Zhao et al., 2010; Wu et al., 2012; Guglielmin and 42 Vieira, 2014).

Aside from the climate and human activities, changes in the active layer are strongly linked 44 to factors such as the physical and thermal properties of the surface soil, vegetation, soil moisture 45 content, and seasonal snow cover (Brown et al., 2000; Hinkel et al., 2003). Seasonal snow cover 46 has significant and complex effects on the hydrothermal regime of the active layer as a result of its 47 unique thermal properties. The high albedo of snow cover (98%) is helpful for reducing the snow 48 surface temperature. In high latitude areas, the average temperature of the nival surface in winter 49 is 0.5-2.0 °C lower than the air temperature (Weller, 1974; Yershov, 1998). The large latent heat 50 (335 kJ/kg) delays the snow cover thawing process and the ground heating rate by a significant 51 amount (Zhang, 2005). In addition, the evaporation of snow meltwater can also help to reduce the 52 land surface temperature. Good thermal insulation occurs in thick layers of snow because of the 53 small thermal conductivity coefficient of snow cover (0.15 W/m·k) (Zhang et al., 1996). However, 54 the thermal conductivity coefficient of snow cover is not fixed (Sturm et al., 1997). Monitoring 55 results from the Alps indicate that the increase rate of the snow cover thermal conductivity 56 coefficient is 0.01 W/m·k·d (Morin et al., 2010). A remarkable increase in this value, even by an 57 order of magnitude (Reimer, 1980), can be caused by the wind (Yen, 1965).

Dramatic spatio-temporal differences in the effects of snow cover on the active layer have 59 been observed due to the thermal properties mentioned above (Zhang, 2005). In high latitude areas 60 with thick snow cover, the temperature of both the active layer underneath the snow cover and the 61 permafrost is often significantly higher than that of bare land, with a 20 °C temperature difference 62 in some areas (Smith, 1975). In Alaska, ground temperatures at depths of 0.29 m and 3.0 m 63 dropped by 1.48 °C and 0.72 °C, respectively, when the snow cover thickness reduced from 40 cm 64 to 20 cm (Ling and Zhang, 2006). Daniel (2001) discovered that snow cover with a thickness 65 greater than 80 cm will have remarkable thermal insulation, and a decrease of 10 cm in snow 66 cover thickness can reduce the mean annual ground temperature (MAGT) by 0.3 °C. In the Amur 67 region of the Greater Khingan Mountains, snow cover 21-36 cm thick can increase the mean 68 annual ground surface temperature (MAGST) by 2.8-5.0 °C (Liang et al., 1993). In the Altai 69 Mountains in northwestern China, seasonal snow cover increases the temperature difference 70 between the ground surface and the atmosphere, which reaches 4.6-7.0 °C in the lower mountain 71 belt and 10°C in the medium mountain belt (Tong et al., 1986).In contrast with the thermal 72 insulation generally discovered in the Arctic Pole and the subarctic region, the effects of snow 73 cover on active layer temperature in the Antarctic Pole and mid-latitude regions are linked to snow 74 cover thickness. In the Antarctic continent, a cooling effect was observed when the snow cover

thickness was less than 0.6 m (Goyanes et al., 2014; Guglielmin et al., 2014). In mid-latitude areas 76 of the Alps, results from bottom temperature of snow (BTS) measurements indicate that 0.8 m is 77 the critical thickness for thermal insulation of the snow cover (Keller and Gubler, 1993), while 78 numerical simulation results show a critical thickness of 0.6 m (Luetschg et al., 2008). Jin et al 79 (2008) analyzed previous research data and proposed that, in eastern parts of the QTP, thermal 80 insulation occurs in seasonal snow cover when its thickness is more than 20 cm, which is similar 81 to monitoring results from the Qilian Mountain ice groove (Hao et al., 2009) and predictions using 82 the Coupmodel (Zhou et al., 2013). In addition, snow cover formation and thawing time can also 83 deeply influence the active layer temperature. Daniel (2001)analyzed the thermal regime of the 84 active layer over the Corvatsch site in the Alps and found that snow cover 5-15 cm thick in late 85 autumn could more effectively cool the shallow soil mass.

Snow cover influences not only the temperature and thickness of the active layer, but also the 87 soil moisture content. In spring, water content in the active layer increases remarkably, even 88 reaching saturation conditions, because of the infiltration of melted snow (Hinzman et al., 1991; 89 Hinkel et al., 2001). In winter, the permafrost shell thickness of the surface layer significantly 90 influences the infiltration of melted snow, while a permafrost shell more than 0.4 m thick could 91 impede infiltration (Iwata et al., 2011). Using observation results from high latitude areas, the 92 SNOW-17 snow cover energy and water balance model has been developed, which theoretically 93 discusses the effects of seasonal snow cover on the water content of the active layer (Anderson, 94 1976).

Previous studies have shown that seasonal snow cover remarkably influences the 96 hydrothermal regime of the active layer, producing significant spatio-temporal differences. In this 97 study, the western section of the Qilian Mountains in the northeastern QTP is investigated, where 98 mountain island permafrost dominates (Li et al., 2012; Li et al., 2014), and a wide distribution of 99 snow cover exists (Zeng et al., 1985; Chen et al., 1991). During the period from 2003 to 2010, 100 there has been a remarkable decrease in the number of average snow days and a gradual increase in the stable snow cover (Sun et al., 2014). Because of differences in geographical location, the 101 102 area in this study differs significantly from the more commonly studied high latitude and Alpine 103 regions with respect to radiation, climate, and snow cover characteristics. Recent studies on snow 104 cover effects on the active layer in this area have mainly focused on numerical simulations and the 105 shallow soil layer at a depth of about 50 cm (Jin et al., 2008; Wang et al., 2011; Zhou et al., 2013; 106 Xiang et al., 2013). As the active layer thickness of the Qinghai-Tibet Plateau is usually 2-3 m 107 (Wu and Zhang, 2010), it is very difficult to objectively evaluate the effects of seasonal snow 108 cover on the hydrothermal regime of the active layer in this area without deep hydrothermal 109 monitoring.

## 110 2. Description of Monitoring Site and Equipment

### 111 **2.1 Description of monitoring site**

The monitoring snow site, including the NSS and SRS, is located in the Yashatu basin of the western Qilian Mountains in the northeast of Qinghai-Tibet Plateau, about 80 km from Delingha city, Haixi Prefecture, Qinghai Province in the southeast and about 30 km from the Qaidam Basin margin in the south, at 96.516° E and 37.6952° N (Fig.1a, b). The average altitude of this site is approximately 4040 m. The snow site and its surrounding areas are flat with a maximal gradient of 0.5°. The Zongwulong Mountain, which runs nearly east to west at an altitude of 4500 m, is located between the Yashatu basin where the snow site lies and the Qaidam Basin. There is a

- remarkable difference in the climate of Yashatu and Delingha city, attributed to influences of the
- Zongwulong Mountain and the altitude contrast.
- 121

a. Site location