# Peer review of "Effects of Seasonal Snow Cover on Hydrothermal Conditions of the Active Layer in"

_The Cryosphere, 2016_

## Referee Comment (RC1) · A. Rist (Referee) · 4 Aug 2016

General comments:

This discussion paper addresses the interesting and for geocryology important topic of the snow cover's effect on the hydrothermal regime of the active layer. While this topic was already intensively studied in Arctic and Alpine permafrost regions, little attention was paid to it in a high-altitude arid steppe underlain by permafrost such as the Qinghai-Tibet Plateau.

The literature review given in the introduction is quite comprehensive for a paper. It could be sharpened by reducing the number of mentioned studies to those which can

be clearly and closely related to the present study.

The objectives or research questions of the study should be expressed more clearly. The applied methods are suitable to achieve the objectives (as indirectly derived from the introduction). One problem is that not all parameters were measured for the same period of time which makes comparisons and correlations difficult. It is in the nature of field research that improvement opportunities for methods only shape out during the measurements and that the analysis of the gained data can require to measure additional parameters or to apply further methods. However, to make the paper better understandable I suggest to give an overview in a figure or table when which parameter was measured.

The results are overall structured and presented logically. Some information given in the results would fit better under the methods (see specific comments). Single argumentations in the results and discussion are partly not easy to follow for the reader and could thus be presented more consecutive and structured. The discussion is structured by questions used as headings. This is an elegant way to show concisely what the reader can expect from this chapter. One subchapter is dedicated to the thermal and hydrological effects of the snow cover on the active layer, respectively. However, to discuss the interrelations between the thermal and hydrological effects would be interesting and complete the discussion. The extension of the discussion by these interrelations could be compensated by restricting the contents to those which are clearly and closely related to the topic of this paper.

The conclusions comprise the study's findings concerning the snow regime, its effects on the thermal and hydrological regime of the active layer, respectively. Also here a conclusion relating the heat and water balance being influenced by the snow cover would be desirable.

Concerning the English language I can neither assess nor give advice for improvements as I am not a native speaker as well. However, I recommend to ask a native
speaker to smoothen the English.

Although I checked minor revisions my comments are quite numerous. However, they are not severe concerning the content. After the revision I strongly recommend to accept this paper for publication in the journal TC.

Specific comments:

Line 13: Make transition smoother from first to second sentence;

Line 17: An event cannot be thick –> find correct formulation;

Line 19: Not clear; could it be like this: cooling of the active layer will be increased if the snow is removed all year round and even more if it is removed in winter, spring and autumn, but left in summer;

Line 21: Don't understand; was the snow removed only some weeks in autumn and then left? Above you wrote that the snow was removed the whole year or at least whole seasons;

Lines 22, 87, . . .: I suggest either soil moisture or soil water content, but not soil moisture content;

Line 24: You already mentioned the decrease of the soil water content in the last sentence of (2);

Line 25: Less compared to what?

Lines 26-28: First mention that you observed lower soil temperatures and higher soil water contents at the NSS than at the SRS; then come up with your interpretation; another reason is the higher albedo of snow; look for literature on this topic;

Lines 36-37: Do all these authors in brackets give statements to all topics you mentioned in the sentence before? If not, please put each citation directly after the corresponding topic to make clear which content originates from which author/s;

Line 39: 'Alpine' is an adjective, but can not be used in addition as a noun as 'Arctic' or 'Antarctic'; furthermore, 'Alpine' is usually related to the European Alps, while 'alpine' identifies an altitudinal belt in mountains;

Line 39: Do all these authors in brackets give statements to all topics you mentioned in the sentence before? If not, please put each citation directly after the corresponding topic to make clear which content originates from which author/s;

Line 47: Only for fresh snow; for old snow the albedo can drop below 80%;

Lines 49-50: 'specific latent heat of fusion at the transition from ice to liquid water' rather than 'latent heat' as the released energy is related to a given mass; where did you get the value 335 kJ/kg from?; in all physics books I know it is 334 kJ/kg;

Line 50: Delays compared to what? To an imaginary lower value of the specific latent heat of fusion?

Lines 53, 54: Only 'snow' instead of 'snow cover' because a cover has a thickness but here you refer to a material constant;

Line 55: Do you mean the European Alps? there are also other Alps like those e.g. in New Zealand;

Line 55: I suggest '... that the daily rate of the increase in the thermal conductivity coefficient of snow is ...'; I guess this increase takes place during the melting period, but you have to tell the reader otherwise it is not clear.

Line 67: Name the country where these mountains are;

Line 69: Regarding 'temperature difference': as average over the whole winter season?

Line 72: Why 'Pole'? Do you mean ....'in the arctic and subarctic regions'?

Line 77: Well, there is also an insulation effect below the given values, which rather refer to a (nearly) complete thermal insulation of the ground from the atmosphere by

the snow cover;

Line 85: More than what?

Line 89: 'the permafrost shell thickness of the surface layer' –> I don't understand what you mean by this; please highlight;

Lines 91-94: What is the main finding of this study related to your study?

Lines 95-96: You already explicated this statement in lines 43-59; avoid repetitions; you could start the sentence like: While previous studies investigated ..., in this study ...';

Line 98: Not 'mountain island permafrost' but 'patchy mountain permafrost';

Line 100: Please explain these findings more detailed; Lines 101-109: Clearly tell the objectives or research questions of your study somewhere after the literature review and the gaps of knowledge;

Line 105: Do you mean '... down to a depth of ...?';

Line 110: I suggest simply: '2. Methods';

Line 111: 'Description of' is dispensable, omit it;

Line 121: Label the figure parts using small letters (a, b, c), refer to them in the figure caption (below the figure) an give explanations only there;

Line 122 (Fig. 1a): Red points for cities are quite dominant compared to the landscape and especially the monitoring site; I recommend to make at least the monitoring site more visible (e.g. red frame);

Line 122 (Fig. 1a): The unit is usually given in [...]; would also do it like this here and throughout the manuscript (please check in all figures, tables and in the text where you give the unit after a numerical property);

Line 122 (Fig. 1a): I recommend to use the color gradient for the elevation inversely:

green for lowest and brown for highest altitudes (more common and intuitive);

Line 122 (Fig. 1a): Add a scale bar;

Line 124 (Fig. 1b): Could you show the topographic map or an aerial photograph in the background to get a better impression of the soil surface coverage?

Line 124 (Fig. 1b, Legend): 'Monitoring site' is exactly the same expression as for the whole monitoring site you show in Fig. 1a - this is misleading; I suggest to use the word 'monitoring plot' for the areas you called NSS and SRS which then change to NSP and SRP;

Line 127: Give meaning of all abbreviations; figures and tables (in combination with their caption) have to be understandable by its own; check for abbreviations in figures and tables throughout the manuscript;

Lines 130-131: I suggest '..., the air temperature in the region ranged between minimum values of -32.6 °C and maximum values of 22.9 °C while the average annual value was ...';

Line 131: Why do you give a range for the average annual values but not for the minimum and maximum annual values?

Line 134: I suggest: 'mean annual gust speed (measured half-hourly)';

Line 134: What is the name of the river? Also show it in Fig. 1 a, b;

Lines 148 and 150 (Fig. 2a and 2b): 'Total depth' means what? Depth of the borehole, 6 m?

Lines 148 and 150 (Fig. 2a and 2b): Regarding 'Column 1:100' –> On my screen (at a view size of 100%) 1 m is equivalent to 6 mm, so the scale 1:100 might be correct in the original figure but it was changed when implanting in this document;

Lines 148 (Fig. 2a): Regarding second layer from top –> I can hardly believe that the

soil can be saturated at a water content of 2.6 % (at 2.1 m depth) within this layer which you generally call water saturated (equivalent for third layer from top in Fig. 2b);

Line 153 . . .: You are describing the equipment here but the heading of 2.1 only refers to the monitoring site;

Line 157: Here you just refer to NSS and SRS. Explain your concept of investigating the effect of the snow cover by comparing one site with a natural snow cover with another one where you artificially removed the snow; this is essential for the understanding of the whole study; and describe it before you come to the measurement equipment;

Line 160: Ground temperature at which depth?

Line 160-161: Do you mean: maximum duration of the snow season?

Line 162: In line 153 you already mentioned a ground temperature equipment; how is it related to the probe you mention here?

Line 167: Why did you measure the soil thermal flux only at the NSS, but not at the SRS?

Line 170 (Table 1): Here you list the measurement parameters wind speed, rel. air humidity, air pressure, precipitation, albedo in the methods but you don't present them in the results or discussion; If you don't need them for your paper delete them in Tab. 1;

Line 170 (Table 1): As not all parameters were monitored for the same period of time it would be helpful to give the monitoring period for each parameter, either in this table or even more vivid in a figure;

Line 170 (Table 1): To me the following order of columns from left to right would be understandable more intuitively: Type of probe, Number of probes (NSS [as reference first], SRS), Measuring range, Measuring accuracy, Brand, Model;

Line 170 (Table 1): Please tell the measuring interval(s) of the different parameters;

Line 170 (Table 1): Why did you use different soil moisture and albedo probes at SRS and NSS? This makes it more difficult to compare the measured values;

Line 170 (Table 1): for the soil moisture probes you have to say if % refers to the gravimetric water content (as in Fig. 2) or the volumetric one;

Line 175: Not clear. Do you mean you used the measured air temperature to correct the signal of the ultrasonic sensor measuring snow depth?

Line 180: All means all, so you don't need to list some probes;

Line 182: Specify the logger by the brand, at least;

Line 186: But I can see a fence around the SRS in Fig. 3b and c; so what do you mean by snow fence?

Line 188: How could you guarantee that the SRS wasn't covered by snow again due to wind after snow removal?

Line 193 . . .: A lot of information on the acquisition of soil temperature data was already given under 2.1; the information given here would fit to table 1; temperature measurement is a standard technique; why do you describe it here so extensively?

Line 205: See comments on the heading of 2.2 (Line 193) and transfer to 2.3;

Line 212-213: Did you dig the test pit and push the rods of the probes horizontally into a side wall or did you burry the probes in layers?

Line 222: Repetition, information already in Tab. 1;

Line 223: What do you mean by hydrothermal probes? The soil temperature and moisture probes together? Did you have to dig into the active layer to install the mast onto which the ultrasonic snow depth sensors is mounted or why do you mention digging in connection with the snow depth measurements?

Line 228: Do you mean because of the calibration for unfrozen conditions by the manufacturer? How did you get the true value? By gravimetric water content measurements using soil probes? Or do you stress the fact that the probes measure the liquid water content but not the frozen water content? Not clear;

Line 237: Do you mean accumulated in total?

Line 245: I question if the snow depth can be determined with a resolution of 1 mm (surface hoar on the snow can be thicker than 1 mm);

Lines 251-252: '... snow cover remained 5 days at maximum in more than 90 % of the snowfall events, ...';

Line 253: In the same sentence you say that the snow cover would have been lasting not more than 5 days in general - this is a contradiction; instead of the words 'generally' and 'typically' try to quantify in the statements;

Line 270 (Tab. 2): Data of air temperature and radiation averaged over the seasons would be helpful here as you argue in the text with these parameters;

Line 272: As you define the depth negative in Fig. 7 the maximum depth would be 0; so either you can say something like 'the most negative depth' or you use a positive sign for depth values in Fig. 7;

Line 277: I question if the ALT can be determined with a resolution of 1 mm;

Lines 284-288: This sentence needs to be related to the ground temperature measurements of this study;

Line 286: Regarding 'at any depth' –> the mean annual ground (rather than soil) temperature MAGT refers to the depth of the zero annual amplitude (see Wu and Zhang, 2008, caption of Table 2);

Line 291: Do you mean the extremes of daily air temperatures within a year? This must become more clear because the extremes of the air temperature within a day must not be excluded to estimate the daily geothermal propagation depth;

293: Did you determine the daily geothermal propagation depth? If not, please cite the publication where you got this information from;

Line 294: It depends on the purpose; what do you want to say with these data?

Lines 293-296: I understand the sentence up to '... single time', but the words thereafter (several times, even partial time of each day) I can't logically relate to the rest of the sentence. Maybe 'or' is missing before 'even'?

Line 303 (Fig. 8): The colors in Fig. 8 should be harmonized with those in Fig. 10 regarding their meaning;

Line 303 (Fig. 8f): How do you interprete that the MAGT-curves are bent towards lower temperatures above 1.5 m? Compare to Fig. 3 in Smith & Riseborough (2002): Climate and the limits of permafrost;

Line 303 (Fig. 8f): I recommend not to choose blue and red lines in Fig. 8d as these colors are used in a different meaning in Fig. 8a - 8e;

Lines 304-307: In line 304 you say 'annual' for the entire Fig. 8, but later you say that the temperature profiles are averaged over seasons!

Lines 309-311: This should additionally be described in methods;

Line 323: 'In terms of yearly temperature,' is dispensable;

Lines 323-325: As first sentence of this paragraph I suggest: 'The mean annual soil temperature at 0.5 m depth was 0.8 °C higher for the NSS than the SRS while the temperature difference between the two sites decreased with depth being approximately zero below 2 m depth.';

Lines 325-326: According to Figure 8e they are approximately the same compared to the layer above 1.6 m!

Line 327: What do you mean by 'generally' here? Average difference over the whole

depth profile or only below 1.6 m? Not clear;

Lines 328-329: This sentence in the present form belongs to methods but not to results; don't repeat methods in results;

Lines 323-333: Not clear! I suggest: 'The MAAT in Yashatu was -4.5 °C, -3.4 °C, and -3.9 °C in 2012, 2013 and 2014, respectively, indicating ...';

Line 333: Where did you get the value for 2012 from? According to Fig. 4 you only measured the air temperature since December 2012; you have to cite the source if it is not your own measurement;

Lines 338-339: This sentence is only valid for NSS, tell the reader;

Lines 340-341: Regarding '..., there is ...' –> Where? Do you mean at the SRS? So you have to tell it the reader. However, I see something different in Fig. 9: besides the small L-shaped area at SRS with a water content above 40 % in 2013 the water contents at NSS are higher throughout the depth profile and throughout the monitoring period;

Line 342: 'based on CS616' does not belong to results but to methods and shouldn't be repeated;

Line 342: It can, but does it or does it not?

Lines 343-345: There appeared strong changes during redistribution, but if you direclty compare the water content profile at NSS in Oct. 2013 and Oct. 2014 I roughly agree although I can see an increase from 30 to 40% between about 1.1 m and 1.7 m depth –> has to be described more clearly;

Line 345: Is the class with the highest water content from 40 to 70%? Please indicate in the color bar in Fig. 9. However, a vol. water content of 70% in gravel (as stated in Fig. 2) is very unlikely to me even for saturation (maybe in clay it would be possible);

Line 349: Just to describe what you can see anyway in Fig. 10 doesn't give an addi-

tional benefit; add information in the text you can't directly see in a Fig. itself;

Lines 354-355: Sentence is too complicated to express the facts;

Line 356: Give a depth range as in Fig. 10;

Line 358: 'first' and 'then' means from 2012-2013 and from 2013-2014, respectively; express it more clearly;

Line 359 (Fig. 9): What is the upper limit of the highest class above 40 %? Please, indicate on the color bar;

Line 359 (Fig. 9): How do you explain the pronounced steps in the water content profiles, especially in 2013?

Line 360 (Fig. 9, caption): ... 'based on CS616' is an information regarding the methods and must not be repeated here;

Line 360 (Fig. 9, caption): In Table 1 you say that soil moisture at NSS was measured using a probe called SM300 but not CS616;

Lines 364-369: Too complicate; just say that you linearly interpolate between the point measurements using the soil moisture probes;

Lines 364-379: This information would be more adequate in the methods;

Line 379: A range is from a minimum to a maximum value; here you mean the difference;

Line 382: Not 'in 2013 and 2014' but 'from 2012 to 2013 and from 2013 to 2014';

Line 385: Mark figure parts either with a and b and refer to them in the caption or with NSS and SRS but not both;

Line 394: See comment on line 47 and 51;

Line 397: Cite these studies;

Line 399: It did in case of ASR-1, but not for ASR-2;

Line 402: To see the effect of the snow cover at the SRS by comparing the 3 years 2012, 2013 and 2014 (i.e. BSR, ASR-1, ASR-2) is hardly possible as you have 3 samples (years) only, but 2 factors (air temperature and snow cover);

Lines 400-403: You cannot directly compare ASR-2 (i.e. SRS 2 years after snow removal) with NSS because the observation period of ASR-2 was 2013.12-2014.11 (line 311) while it was 2014.3.1-2015.2.28 (line 300) for NSS; however, you can compare SRS and NSS in the same period of monitoring as shown in Fig. 8e. Maybe you meant this, but it is not clear;

Line 407: Are these results yours? If not, cite the publication in the same sentence;

Line 409: This finding means that the ground cooling effect due to the change of the albedo by the first snow cover in autumn is less effective than the cooling due to the stronger heat dissipation without snow; finally, the thermal insulation (and during this season warming) effect of the snow cover overbalanced the cooling effect due to the higher albedo of snow in this study;

Lines 414-415: '... when the ground temperature is higher than the air temperature'?

Line 418: Table 3 shows results and should thus be presented in chapter 3, but not only in the discussion;

Line 418 (Table 3, caption): This is again another period of time than for the collection of the other data (soil temperature, air temperature, water content); so it is difficult to relate them to each other;

Line 424: Which areas?

Line 426: not correct: latent heat is released by freezing and by condensation, but not by a decrease in the water content; if the soil water content in autumn is less than the year before, also the latent heat released during freezing will be less; is it this point you

wanted to make? Then make it clearer, please;

Line 427: See comment on line 50;

Line 429: But at 4000 m a.s.l. altitude the pressure is much lower –> adapt the following calculations to a realistic atmospheric pressure at 4000 m a.s.l.;

Line 431: How did you determine this heat capacity? Please show that it is realistic using the content of mineral material, ice and water;

Line 430-433: Has to be explained more clearly. Just by reducing the water content the temperature won't decrease! However, the heat to be extracted from 1 m3 of soil to freeze the water equivalent to a VWC = 1% is 3350 kJ. If the same heat would be extracted from the same body of soil without freezing (i.e. if already al the water is frozen or al the water remains liquid) a temperature decrease of 1.5 °C would occur. For the heat of vaporization the argumentation is equivalent;

Line 435 . . .: I can't see that you used the thermal flux data to verify 'this phenomenon' explained above. I would just argue that at higher water contents more heat has to be extracted for freezing than in drier soils. The atmosphere can take up only a given amount of heat under given meteorological conditions. After freezing, in driers soils more heat that can be dissipated is left for ground cooling resulting in lower temperatures;

Line 438: To make these four stages more visible I suggest a diagram rather than a table;

Line 444: It should be like this, but Table 3 shows the opposite! You argued for all other months. I would say the higher/lower values in February/June at the NSS than at SRS is just natural variability;

Line 446: Even greater than in Table 3? And why should the heat exchange be different at NSS and SRS if there is no snow? Because of the higher water content at NSS than at SRS? Then you need to argue which processes lead to the assumed result; however,

it has to be consistent with your other findings;

Line 456: Instead of the grain size distribution it is rather the pore size distribution;

Line 457: Why and how? I would say the pore size distribution, the porosity and thus the bulk density are changed by digging, but not the grain size distribution;

Line 462: Is it possible that the digging lead to preferential flow paths in the ground? However, you dug at both sites in the same way, right?

Lines 471-473: Does the annual rainfall of Delingha originate from 1960? Or was it only published then and is a long term mean, i.e. even older? It could have changed a lot since then! So you can't compare it with todays values of Yashatu;

Lines 480-481: Is half a page really necessary to come to this explanation?

Lines 483: Repetition of line 469 –> delete;

Line 483: 'the melt water equivalent to the SWE' instead of 'this result';

Line 483: '... could have increased ... by only 4.4 % ...';

Line 493: Only in summer, when the evaporation is highest, the NSS was cooler than the SRS at 0.5 m depth (Fig. 8). However, then there was nearly no snow (Table 2). So I don't think that this effect contributed to the lower water content at SRS, otherwise the near surface temperatures (0.5 m depth) should have been higher at SRS than at NSS also in spring, winter and autumn which was not the case;

Line 497: But according to Tab. 2 there was nearly no snow in summer at your site;

Lines 498-499: But in Fig. 8 you have shown that the mean winter temperature is lower at the snow free SRS than at the snow covered NSS!

Line 504: Also this sounds logical but should have resulted in lower near surface temperatures at NSS than at SRS, also in winter, spring and autumn. Or do you think this was the case, but you couldn't show it as no data could be gained above 0.5 m depth?

Then you have to say it clearly;

Line 507: Which range do you mean - within a year? Then during the first year the VWC ranged between 0 and the highest class (40 - ?%), in the second year between 0 and the second highest class (30% - 40%). So what do the 50 % decrease mean?

Line 508: '...as the snow removal duration increases' –> does it mean 'as long as the snow removal will be continued'?

Line 515: The mean surface?

Lines 523-525: The topic of your paper are the effects of the seasonal snow cover on the hydrothermal conditions of the active layer; so here you should conclude that the snow removal at SRS lead to lower water contents which can be derived from the comparison with the NSS where the measurements took place at the same period of time facing the same (dry) meteorological conditions. So far (in point 3 of the conclusions) you argue only by the temporal sequence of 3 years;

Technical corrections:

Line 15: Replace '.' by ':';

Line 21: Instead of 'maybe results' use 'may result';

Line 36: Add space between 'Brown' and 'et', and 'Lemke' and 'et;'

Lines 47, 51: Avoid words like 'helpful' or 'help'; they indicate that you want something to be or not to be; however, as scientists there is no good or bad, you should just investigate nature without judging; otherwise you are subjective rather than objective;

Line 48: Why not simply 'snow surface'?

Line 52: A physical property can't be good or bad;

Line 53: The K has to be big as it means Kelvin but not thousand;

Line 56: K instead of k in the unit W/(m·K·d);

Line 59: Add a space after the fullstop;

Line 60: '... with a thick ...'

Line 61: '... with a temperature difference of 20 °C ...';

Line 64: '... that a snow cover ...';

Line 65: not 'will have' but 'has';

Line 65: '... a remarkable ...';

Line 67: I am not a native speaker, but the English in this sentence doesn't seem to be correct to me; check with a native speaker, please;

Line 69: '..., the seasonal snow cover ...';

Line 71: Add space between 10 and °C;

Line 71: Add space after fullstop;

Line 74: Isn't it 'on the continent'? check with a native speaker, please;

Line 78: Add a point after 'al';

Line 82: Isn't it written CoupModel?

Line 83: I suggest 'strongly' rather than 'deeply'; check with native speaker;

Line 83: Add space after (2001);

Line 84: 'at' instead of 'over';

Line 84: '... the European Alps ...';

Line 84: '... that a 5-15 cm thick snow cover ...';

Line 88: Instead of 'melted snow' I suggest 'snow melt water'; besides the melt water originating from ground ice;

Line 92: Put the acronym 'SNOW-17' after 'snow cover energy ... model';

Line 126 (Fig. 1d): Picture is slightly higher than that of Fig. 1c –> adjust

Line 135: '... found (Figure 1d)';

Line 136: ' ... 20% (Figure 1c). ';

Line 139: '... at the snow site ...';

Line 140: 'Mudstone occurs at a depth of 5.0 m for the NSS and 3.6 m for teh SRS (Figure 2)';

Line 144: '...-0.32 to -0.30 °C...'; then use 'to' to indicate a range throughout the document;

Lines 148 and 150 (Fig. 2a and 2b): To make it clearer I recommend to omit the vertical lines between a property and its value, e.g. Altitude: 4040 m;

Lines 148 and 150 (Fig. 2a and 2b): Regarding column 'Notes' –> To save space in the figure itself you could give this information in the caption;

Line 151: Mark figure parts either with a and b and refer to them in the caption or with NSS and SRS but not both;

Line 153: September 2009?

Line 154: '... meteorological measurement equipment ...'

Line 154: November 2009?

Line 155: '... monitored using the sensors ...';

Line 157: '. . . in May 2010 . . .'

Line 162: '. . . in May 2010.';

Lines 162-163: 'A set of ... was installed ... of the two plots, NSS and SRS, respectively.'

Lines 165-168: '... were added at the NSS and the SRS, while an ultrasonic snow depth sensor and a sensor measuring the soil thermal flux at shallow depths were installed only at the NSS.';

Line 170 (Table 1): Use small letter at the beginning; also for soil temperature;

Line 170 (Table 1): Under the name 'types' you are mixing parameters (e.g. AT&H) and measuring intsruments (e.g. barometer or rain gauge) –> harmonize, please; use singular, e.g. 'Type of probe';

Line 170 (Table 1): 'Measuring range' instead of 'Ranges';

Line 170 (Table 1): 'Number of probes' instead of 'Numbers';

Line 170 (Table 1): In order to save space you could use small superscript numbers directly after the probe type;

Line 177: Add space: 6 cm;

Line 178: '... are thus needed ...'; then you can omit: 'because ... rapidly';

Line 182: '... to the automatic data logger CR3000.';

Line 183: instead of 'that ... when' use 'after';

Line 184: '... station, data were often not recorded during night at the SRS.';

Line 185: '... area differing between seasons.';

Line 191: Distance between the three pictures are slightly not the same as well as their height –> adjust;

Line 193: 'soil temperature data acquisition' or 'soil temperature measurement';

Line 207: '... two parallel steel rods which are 300 mm long, 3.2 mm in diameter and separated by a distance of 32 mm.';

Line 213: Not 'laid by drilling' but 'installed in a bore hole';

Line 214: 'installed' instead of 'laid';

Line 214: '... reaches its maximum thawing penetration.';

Line 216: 'or' instead of 'and';

Line 217: '... were thus due ...';

Line 219: '... extended from the ground surface to 3.6 m depth?';

Line 220: Not 'laid' but 'installed' (2 times in this line);

Line 224: Not 'by' but 'in'

Line 224: Instead of 'wouldn't' I would say 'shouldn't';

Line 226: '... water content in the thawed soil ...'?

Line 237: Why do you say 'surface snow cover' instead of simply 'snow cover'? As the snow cover is always deposited on the surface this word is dispensable;

Line 239 (Firg. 4): Curves are too thick (only in pdf?); add horizontal lines as visual orientation;

Lines 246, 248, 252: Use past tense because it occurred in 2014; check whole manuscript and use past tense for all events which were finished in the past;

Line 246: Check with editor if this date format is ok; However, the date format should be the same throughout the document which is not the case, e.g. line 245 (December 2013) differs from line 240 (2012.12);

Line 260: Add space after fullstop;

Line: 277: 'Figure 7' but not 'figure 7';

Line 280 (Fig. 7): Indicate at least one negative isotherm;

Line 280 (Fig. 7): Mark figure parts either with a and b and refer to them in the caption or with NSS and SRS but not both;

Line 290: '... where the daily soil ...';

Line 298: 'could' instead of 'can';

Line 301: 'also' is dispensable;

Lines 329-330: '... removal, the mean annual ground temperature at 0.5 m and 2.0 m depth increased by 0.3 °C and 0.2 °C, respectively.';

Line 334: No comma after 'namely';

Line 349: Don't make a new paragraph after only 1 sentence (minimum after 2);

Lines 350-351: 'At a depth of 0-50 cm soil moisture sites varies no more than 4 % between the years 2012-2014 at each of the two sites.';

Line 351: 'soil moisture decreases with time, and' –> not needed, delete;

Line 354: 'Between 80 cm and 120 cm depth, soil ...';

Line 372: '... at 0-5 cm depth is assumed to be the same as ...';

Line 390: Add space after Alps;

Line 391: Don't use first names for the citations in the text (only initials of first names in bibliography);

Line 401: A comma or 'and' is missing after 'snow removal';

Line 405: '... heat dissipation from the active layer to the atmosphere in winter (...).';

Line 407: Omit 'the';

Line 411: In addition to Fig. 6, also refer to Table 2;

Line 418 (Table 3, caption): The table is obviously not at the right place but should be

positioned later;

Line 421: '... since the snow ...';

Line 423: '... of the active layer.';

Line 429: '... heat for water stored in a 1 m3 soil body at a volumetric water content (VWC) of 1 % is 3350 kJ ...';

Line 437: Add a space before 'According';

Line 464: 1 space before 'therefore' is engough;

Line 469: '...), resulting in a snow water equivalent (SWE) of ...';

Line 484: '... content between 0-2.5 m depth in the active layer ...';

Line 508: Add a space after the fullstop;

Line 524: '... SRS can be attributed to the removal of ...';

Line 628: Check that the authors' family names and the initials are always separated by commas in the bibliography (which is not the case e.g. in line 628 and 630)

---

## Referee Comment (RC2) · K. Gisnås (Referee) · 24 Oct 2016

Introduction:

This paper analyzes the snow cover's effect on the hydrothermal regime of the active layer. In contrast to the majority of previous studies on this topic, this study is conducted in the very cold and dry climate of the Qinghai-Tibet Plateau. The snow cover season is marginal, with snow cover duration normally no longer than 5 days, and snow heights typically 5 cm. To analyze the effect of the snow cover on the ground thermal regime, snow is removed after every snowfall, and ground temperature and soil moisture measurements are compared between this site and a natural site. The authors find that the removal of the marginal snow cover still has an effect on the ground, and

drying and cooling of the ground is observed at the snow removal site.

This study addresses a topic much investigated before. The climate of the study location, however, is very different from the environmental setting of most other studies of this type, making the study both interesting and an important scientific contribution. The very short duration of snow covered ground and very shallow snow cover results in other dominating effects from the snow cover on the underlying ground, and the large amount of field data allows for a detailed analysis of the hydrothermal effects.

Despite this, the paper lacks a logical structure in the argumentation, and it is difficult for the reader to understand the background for the focus of the discussion and the conclusions drawn. The structure and the presentation of the results must be significantly improved, and the results that the statements in the discussion are founded on must be highlighted and presented in a clearer way. The paper in its current form is difficult to evaluate with respect to the scientific content of the discussion and conclusion. For this reason, I recommend a major revision of this paper before publishing. A list of general and more specific comments and suggestions to the paper is provided below.

General comments:

1. The introduction chapter includes a very detailed literature review. I recommend to shorten this part, and only include the background necessary to put the paper in a larger context. Highlight why this study is unique and needed in context of previous studies on the same topic in the introduction, but avoid starting the discussion here. Rather move parts of it (with many of the references) to the discussion chapter where you discuss the results in relation to previous findings.

2. I miss a presentation of the objectives in the paper. Please include clear objectives, e.g. in the last paragraph of the introduction chapter.

3. The different observation periods for each variable is confusing for the reader. Please make a figure or table illustrating the period of measurement for each vari-

able/instrument, e.g. as a timeline of observations. In addition, I cannot find explicit information on the period when the snow removal is done. I assume this was done for the entire period 2012 to 2015? This is crucial information in this paper! Please specify.

4. The authors state that the temperatures at NSS are warmer than at SRS over a calendar year, and suggest that removal of the snow has a cooling effect on the ground. However, I miss a clear quantification of the difference between the sites and how this changes with time, supporting these statements. Does the difference increase by each year? Functions of running mean annual temperatures of some selected temperature loggers (depths) would be useful, as well as MAGT for each year at each site.

5. How can the ALT be determined with an accuracy of mm in the range 3.4 - 3.6 m, when the soil temperature measurements are only located at 3 m and 4 m depth? These depths are derived from (I assume linear?) interpolation of the temperature logger data. Because of variation in ice content and ground material this may not be entirely true, and the use of mm precision does not make sense. The ALT derived from interpolation can therefore not be used to differentiate the change in ALT between the two sites. I would say it could perhaps give an indication of ALT thickness within 10 cm, but it has to be noted in the paper that this is an approximation. By this you could still say that ALT at both sites are increasing, but you cannot differentiate the ALT change. In order to assess the differences between the sites, please compare observed temperatures at 3 and 4 m depths between the sites.

6. The actual effect of the snow removal on ground temperatures is not clear to the reader (see points 5 and 6). It is therefore also difficult to follow the discussion of why snow removal has a cooling effect. However, IF the effect is cooling at SRS compared to NSS, the discussion must focus on establishing the cause of this effect. Is the reason a change in thermal insulation, albedo, efficiency of longwave radiation exchange, energy lost to snow melt or infiltration of meltwater/soil moisture (see e.g. summary in Zhang, 2005)? In most areas with a developed snow cover the first effect (thermal insulation) would dominate, and the result of snow removal would be cooling of the ground. However, as the authors correctly highlight, a 5 cm thick snow cover is normally considered too thin to have an insulating effect on the ground. Still, the authors spend most of the paper discussing whether the thermal insulation from snow is the reason for the cooling effect. As a reader, I would really doubt that this is the case, and therefore it is crucial that you support this discussion with observations. The most obvious would be to compare hourly temperature observations from the air to the uppermost logger (5 cm) in both boreholes. In this way you could see if there is a pronounced dampening of the daily temperature amplitude after a snowfall at 5 cm depth at the NSS site, and not at the 5cm observations at the SRS site.

7. The entire discussion and logical structure behind arguments has to be improved throughout the paper. It is difficult for the reader to relate the discussion around effects to the presented results. The soil moisture data presented in Figure 10 shows interesting results, with a general drying of the SRS site. There is rapid decreases at some of the depths, which has to be commented upon. What is the accuracy of these observations? The authors also link the drying of the ground to reduction of melt water infiltration and increase of evaporation at the SRS site. This seems very likely, but I miss the explicit link from this to the thermal effect it would have on the ground. In general, a soil with less moisture would reduce the exchange of latent heat, as latent heat from freezing of soil moisture is a large energy source, while melting of ice is a similarly large energy sink. Over a year the latent heat energy in and out of the system would be equal, if the soil moisture is not changed. However, in this case there is a gradual drying of the ground at the SRS site during the period, potentially resulting in more melting and evaporation than freezing of water during one year. This would be an energy sink and consequently cool the ground. The authors comment both of these effects, which seems very likely in light of the observed soil moisture data. Still, the authors spend more time discussing the effect of thermal insulation in the discussion chapter. This seems strange, as the presented data indicate that the reduced latent heat effect from drying of the soil may be an explanatory factor, and there are no results

demonstrating that the thermal insulation from the snow cover play a major role.

8. In Figure 4 you present the relative humidity observations. However, these are used only to give the average annual relative humidity. It would be nice to connect these observations to the discussion of the effect of increased evaporation in chapter 4.2.

9. The paper would benefit from a better structure and consistency. Some results are mixed into methods chapter (line 138- 146 and line 159-161). The logical structure of the discussion points also has to be improved. Please refer to the results when you discuss them, and ideally quantify the finding, i.e. not only refer to "a cooling effect", but give how much cooling compared to reference site.

10. Please keep a consistent time format throughout the paper. At least four different date-formats are used in figures and in the text. Please clear this up!

11. Figure 4 to 6 and also ideally Figure 7 should be presented together (in the same figure or below each other at one page) with the same date format so that the data can be related to each other! Now they represent different periods of observation, all have different date format, and they are not in the same figure.

12. The text would benefit from a simpler and clearer language. Some sentences are lengthy and could be simplified. This may partly be a language problem, but the general content could also be sharpened.

Specific comments:

Line 21: "Maybe" in the abstract is a bit vague. Rather use "likely", or "we believe". This is also valid for the conclusions (point 3, L 524).

Line 21: What do you mean with "the delay of snowfall time in autumn"? Please clarify, and relate it to the physical process – does this also refer to the insulating effect of snow, or other effects?

Line 53: Change into "Low thermal conductivity of snow". Delete coefficient. What do

you mean with "The Alps"? The Swiss Alps?

Line 58 – 85: Why is the observed effect, given in degrees C, of variation in snow depths higher in some areas than in others? I would expect this to be directly coupled to the climate (very cold winters or maritime mild winters?) Since snow cover with a critical thickness (typically 60-80 cm depending on the snow properties) disconnects the ground surface temperatures from the air temperature due to very low thermal conductivity, it follows that the difference in ground temperatures between a site with 40 cm snow and a site with 80 cm thick snow is closely related to the air temperature. A very cold winter would result in a large difference, while a warmer (maritime) winter with temperatures close to 0°C would result in less difference.

Line 56: Thermal conductivity is normally given as W/m K, or better W m-1 K-1, where Kelvin is denoted with capital K. The latter notation is used later in the paper; please be consistent. Also clarify the meaning of "d" in W/m K d. If this is temporal rate change of thermal conductivity per day, change into W/m K day or W m-1 K-1 day-1.

Line 77: Here I would also refer to Haeberli and the "Bottom temperature of snow" (BTS)-method.

Line 89 and 90: What is the permafrost "shell"? Please clarify.

Line 92-94: I guess there are also several newer models developed for this purpose. Is there a reason why you mention this old one in particular? If not, please remove.

Line 98-99: I don't understand the meaning here. Do you mean "a wide distribution of snow depths"?

Line 100: snow covered days

Line 100-101: gradual increase in the height of the stable snow cover

Figure 1b: This figure does not tell much. Either leave out and give distance between the sites and the elevation in the text, or include some background information on

vegetation type or similar.

Line 131: suggested change: "average annual temperatures"

Line 142: propagation depth at the snow site

Line 138 – 161: The description of the monitoring site (2.1) is quite lengthy. This is partly due to results mixed into the methods chapter (e.g. line 159 – 162). Avoid this and be more concise.

Line 177-178: This sentence does not fit in the methods chapter.

Line 186-188: This is an important part of the method (the removal of snow) which is hidden away.

Line 201 – 208: This information is included in the table, and it is therefore not necessary to repeat it in the text.

Line 274: What is "thawing and freezing process curves"? Please be more precise, e.g.: "continuous plots of interpolated ground temperatures"?

Line 271-278: It is enough to specify how you determine the active layer thickness and the actual thickness; e.g. "Continuous plot of interpolated ground temperatures for the period xx to xx are shown in Figure 7. Here we define the active layer thickness as the maximum depth of the 0 °C isotherm (Muller, 1974). From the continuous plots we find that the ALTs of the two sites are xx cm and xx cm in 2013 and xx and xx in 2014."

Line 291 – 296: This part is unclear. Either refer to observations or cite previous studies.

Line 299: Suggested change: "Profiles of seasonal average soil temperatures interpolated between the loggers from 0.5 m to 4 m depths at SRS and NSS are shown in Figure 8. The averages are made over the period 2014-03-01 – 2015-02-28."

Figure 8: Inclusion of the season (e.g. Mar – May) on each plot would make it easier

to read. Similar x-axis would also make it easier to see the relative variation in temperature differences. In Figure 8f (or in a separate figure) it would be good to also include the evolution of temperatures in the NSS. This would make it easier for the reader to understand if the change in SRS is only due to climatic changes, or if it is due to the removal of snow, and if the initial situation at the two sites where similar or if the variation where as large as in figure 8e also in 2011-2012.

Line 350 – 358: Do you refer to Figure 9 or 10 here? Please specify and make references in the text. Also make sure you highlight interesting points from the figure, and don't reproduce the figure in the text.

Line 371: Do you have a reference on Eq. 1?

Figure 10: Specify that this is soil moisture at maximum thaw penetration (October) each year. Also consider placing this figure together with Figure 9, and indicate the timing of the calculations in Figure 10 with lines in Figure 9.

Line 396-399: Repetition form introduction. Please reduce the amount of redundancy.

Line 399: The ground temperature in the SRS should therefore increase after snow removal.

Line 400 – 403: Repetition. Delete "the thickness of the snow cover was smaller than the critical snow cover thickness" and include "snow removal, while the average soil temperature"

Line 404-405: You state above that thermal insulation from the snowpack is not a dominating effect with snow heights lower than 20 cm. Here you still argument that this could possibly be an effect. Why do you believe so? You have to support this with observations! Again, this can e.g. be done by comparing daily temperature amplitudes in the air and in the topsoil (5 cm depth) before and after snowfall, at the SRS and at the NSS.

Line 407-408: Please include a reference for this statement, or clarify if statement

refers to the same study as referred above.

Line 408 – 417: Is this applicable if the snow cover is NOT thick enough to have an insulating effect?

Table 3: Is snow clearing still in effect in 2015 and 2016? Please be explicit.

Line 434: The "other significant factor" than what? Thermal insulation? There are far more indices that the decrease in soil moisture is an effect than the thermal insulation, which has no effect proven from the data.

Line 435 – 449: Please relate the differences in fluxes to physical processes. Please be more explicit.

---

## Author Comment (AC1) · 15 Jun 2017

**Reply to RC1**

**Replies to the general comments:**

According to your suggestions and new data in 2015 and 2015, the literature is sharpened to one page, the objective is re-expressed, monitoring period of all parameters is pictured and uniformed, the discussions are rewrote, the reciprocal action of hydrothermal effect are added, and the conclusions are promoted greatly.

**Replies to the specific comments and Technical corrections:**

Line 13: Make transition smoother from first to second sentence;

Please see the line 12 to 16.

Line 15: Replace '.' by ':';

Accepted. Please see the line 17.

Line 17: An event cannot be thick –> find correct formulation;

Accepted. Please see the line 20 to 21.

Line 19: Not clear; could it be like this: cooling of the active layer will be increased if the snow is removed all year round and even more if it is removed in winter, spring and autumn, but left in summer;

Revised. Please see the line 21 to 27 and section 2.2.

Line 21: Instead of 'maybe results' use 'may result';

Revised. Please see the line 21 to 26.

Line 21: Don't understand; was the snow removed only some weeks in autumn and then left? Above you wrote that the snow was removed the whole year or at least whole seasons;

Revised. Please see the line 21 to 27.

Lines 22, 87, : : :: I suggest either soil moisture or soil water content, but not soil moisture content;

Accepted.

Line 24: You already mentioned the decrease of the soil water content in the last sentence of (2);

Revised.

Line 25: Less compared to what?

Revised.

Lines 26-28: First mention that you observed lower soil temperatures and higher soil water contents at the NSS than at the SRS; then come up with your interpretation; another reason is the higher albedo of snow; look for literature on this topic;

Revised. Please see the line 17 to 30.

Line 36: Add space between 'Brown' and 'et', and 'Lemke' and 'et;'

Accepted. Please see the line 36.

Lines 36-37: Do all these authors in brackets give statements to all topics you mentioned in the sentence before? If not, please put each citation directly after the corresponding topic to make clear which content originates from which author/s;

Accepted. Please see the line 36 to 39.

Line 39: 'Alpine' is an adjective, but can not be used in addition as a noun as 'Arctic'

or 'Antarctic'; furthermore, 'Alpine' is usually related to the European Alps, while 'alpine' identifies an altitudinal belt in mountains;

Please see line 42.

Line 39: Do all these authors in brackets give statements to all topics you mentioned in the sentence before? If not, please put each citation directly after the corresponding topic to make clear which content originates from which author/s;

Please see the line 41 to 43.

Lines 47, 51: Avoid words like 'helpful' or 'help';

Accepted in the full text.

Line 47: Only for fresh snow; for old snow the albedo can drop below 80%;

It was revised.

Line 48: Why not simply 'snow surface'?

It was revised.

Line 50: Delays compared to what? To an imaginary lower value of the specific latent heat of fusion?

It was revised and deleted in the line 44 to 48.

Lines 49-50: 'specific latent heat of fusion at the transition from ice to liquid water' rather than 'latent heat' as the released energy is related to a given mass; where did you get the value 335 kJ/kg from?; in all physics books I know it is 334 kJ/kg;

Accepted in the full text.

Lines 47, 51: Avoid words like 'helpful' or 'help';

Accepted in the full text.

Line 52: A physical property can't be good or bad;

Yes. It was revise.

Line 53: The K has to be big as it means Kelvin but not thousand;

Yes. It was accepted.

Lines 53, 54: Only 'snow' instead of 'snow cover' because a cover has a thickness but here you refer to a material constant;

Yes. It was revised.

Line 55: Do you mean the European Alps? there are also other Alps like those e.g. in New Zealand;

It was revised and deleted in the line 44 to 48.

Line 55: I suggest '... that the daily rate of the increase in the thermal conductivity coefficient of snow is ...'; I guess this increase takes place during the melting period, but you have to tell the reader otherwise it is not clear.

It was revised and deleted in the line 44 to 48.

Line 56: K instead of k in the unit W/(m_K_d);

It was revised and deleted in the line 44 to 48.

Line 59: Add a space after the fullstop;

Line 60: '... with a thick ...'

It was revised and deleted in the line 49 to 59

Line 61: '... with a temperature difference of 20 _C ...';

It was revised and deleted in the line 44 to 48.

Line 64: '... that a snow cover ...';

It was revised and deleted in the line 44 to 48.

Line 65: not 'will have' but 'has';

It was accepted in the line 53.

Line 65: '... a remarkable ...';

It was accepted in the line 53.

Line 67: Name the country where these mountains are;

It was deleted in the line 49-59.

Line 67: I am not a native speaker, but the English in this sentence doesn't seem to be correct to me; check with a native speaker, please;

It was deleted in the line 49-59.

Line 69: '..., the seasonal snow cover ...';

It was deleted in the line 49-59.

Line 69: Regarding 'temperature difference': as average over the whole winter season?

It was deleted in the line 49-59.

Line 71: Add space between 10 and _C;

It was deleted in the line 49-59.

Line 71: Add space after fullstop;

It was deleted in the line 49-59.

Line 72: Why 'Pole'? Do you mean ....'in the arctic and subarctic regions'?

It was deleted in the line 49-59.

Line 74: Isn't it 'on the continent'? check with a native speaker, please;

It was accepted in the line 55.

Line 77: Well, there is also an insulation effect below the given values, which rather refer to a (nearly) complete thermal insulation of the ground from the atmosphere by the snow cover;

It refers to a nearly complete thermal insulation. Below the snow cover with the thickness of $\geq 0.8$ m, BTS depends on the ground temperature and thus the BTS is often used to judge the permafrost.

Line 78: Add a point after 'al';

It was deleted in the line 49-59.

Line 82: Isn't it written CoupModel?

It was deleted in the line 49-59.

Line 83: I suggest 'strongly' rather than 'deeply'; check with native speaker;

It was deleted in the line 49-59.

Line 83: Add space after (2001);

It was deleted in the line 49-59.

Line 84: 'at' instead of 'over';

It was deleted in the line 49-59.

Line 84: '... the European Alps ...';

It was deleted in the line 49-59.

Line 84: '... that a 5-15 cm thick snow cover ...';

It was deleted in the line 49-59.

Line 85: More than what?

It was deleted in the line 49-59.

Lines 22, 87, : : :: I suggest either soil moisture or soil water content, but not soil moisture

content;

Accepted in the line 61 and the full text.

Line 88: Instead of 'melted snow' I suggest 'snow melt water'; besides the melt water originating from ground ice

Accepted in line 62.

Line 89: 'the permafrost shell thickness of the surface layer' –> I don't understand what you mean by this; please highlight;

It was revised in the line 63-65.

Line 92: Put the acronym 'SNOW-17' after 'snow cover energy ... model';

It was deleted in the line 49-59.

Lines 91-94: What is the main finding of this study related to your study?

It was deleted in the line 60-65.

Lines 95-96: You already explicated this statement in lines 43-59; avoid repetitions; you could start the sentence like: While previous studies investigated ..., in this study ...';

It was deleted in the line 66-74.

Line 98: Not 'mountain island permafrost' but 'patchy mountain permafrost';

It was revised in the line 66-74.

Line 100: Please explain these findings more detailed; Lines 101-109: Clearly tell the objectives or research questions of your study somewhere after the literature review and the gaps of knowledge;

It was deleted in the line 66-74.

Line 105: Do you mean '... down to a depth of ...?';

Yes. It was revised in the line 70-72.

Line 110: I suggest simply: '2. Methods';

Accedped.

Line 111: 'Description of' is dispensable, omit it;

Accepted.

Line 121: Label the figure parts using small letters (a, b, c), refer to them in the figure caption (below the figure) an give explanations only there;

Accepted. Please see the Figure 1.

Line 122 (Fig. 1a): Add a scale bar;

Accepted. Please see the Figure 1.

Line 122 (Fig. 1a): I recommend to use the color gradient for the elevation inversely: green for lowest and brown for highest altitudes (more common and intuitive);

Accepted. Please see the Figure 1.

Line 122 (Fig. 1a): Red points for cities are quite dominant compared to the landscape and especially the monitoring site; I recommend to make at least the monitoring site more visible (e.g. red frame);

Accepted. Please see the Figure 1.

Line 122 (Fig. 1a): The unit is usually given in [...]; would also do it like this here and throughout the manuscript (please check in all figures, tables and in the text where you give the unit after a numerical property);

Accepted. Please see the Figure 1.

Line 124 (Fig. 1b, Legend): 'Monitoring site' is exactly the same expression as for the whole

monitoring site you show in Fig. 1a - this is misleading; I suggest to use the word 'monitoring plot' for the areas you called NSS and SRS which then change to NSP and SRP;

Accepted in the full text.

Line 124 (Fig. 1b): Could you show the topographic map or an aerial photograph in the background to get a better impression of the soil surface coverage?

Accepted. Please see the Figure 1b.

Line 126 (Fig. 1d): Picture is slightly higher than that of Fig. 1c –> adjust

Accepted. Please see the Figure 1c and d.

Line 127: Give meaning of all abbreviations; figures and tables (in combination with their caption) have to be understandable by its own; check for abbreviations in figures and tables throughout the manuscript;

Accepted in the full text.

Line 131: Why do you give a range for the average annual values but not for the minimum and maximum annual values?

I think the both expressions are all ok. Please see the section 3.1.

Lines 130-131: I suggest '..., the air temperature in the region ranged between minimum values of -33.9 ℃ and maximum values of 24.1 ℃ while the average annual value was ...';

Accepted. Please see the section 3.1.

Line 134: I suggest: 'mean annual gust speed (measured half-hourly)';

It was deleted in the revised paper.

Line 134: What is the name of the river? Also show it in Fig. 1 a, b;

This is a seasonal small stream, which can't be found in the local GIS database. It flows away in the northwest of the SRP and the shortest distance between the stream to the SRP is about 200 m. The stream does not appear in the google map (Figure 1b) due to the big distance.

Line 135: '... found (Figure 1d)';

Added.

Line 136: ' ... 20% (Figure 1c). ';

Please see the line 89 in the section 2.1.

Line 139: '... at the snow site ...';

Please see the line 98-99..

Line 140: 'Mudstone occurs at a depth of 5.0 m for the NSS and 3.6 m for the SRS(Figure 2)';

Please see the line 100.

Line 144: '...-0.32 to -0.30 _C...'; then use  Žto' to indicate a range throughout the document;

It was deleted in the text.

Lines 148 and 150 (Fig. 2a and 2b): To make it clearer I recommend to omit the vertical lines between a property and its value, e.g. Altitude: 4040 m;

Accepted in the Fig. 2a and 2b.

Lines 148 and 150 (Fig. 2a and 2b): Regarding column 'Notes' –> To save space in the figure itself you could give this information in the caption;

Accepted in the Fig. 2a and 2b. Contents in the 'notes' are placed in the caption.

Lines 148 (Fig. 2a): Regarding second layer from top –> I can hardly believe that the soil can be saturated at a water content of 2.6 % (at 2.1 m depth) within this layer which you generally call water saturated (equivalent for third layer from top in Fig. 2b);

Reply: Yes, a water content of 2.6 % shows that the soil is unsaturated. This W.C. value may

be connected with the water loss of the drill core. Water retention ability of sandy-gravelly soil, located in the second layer (Fig.2a), is very inferior. During the process of collecting the core in September 2009, the active layer was thawed and water flow could be seen along the core barrel. In the early plan, test pits were expected to be excavated in November 2009. Because the ground water was found at the depth of 0.6 m in the NSP and 0.8 m in the SRP, construction time of test pits were postponed to May 2010. So, the above mentioned soil layer was regarded as saturation in the both figures. To avoid misunderstanding, three W.C. values at the depths of 1.0 m, 1.5 m and 2.1 m, are all deleted in the Fig. 2a. Saturated water content of coarse sand is generally approximately 10.0% The W.C. values at the depths of 2.6 m in the Fig. 2a and 1.5 m in the fig. 2b are still retained in the both figures.

Lines 148 and 150 (Fig. 2a and 2b): Regarding 'Column 1:100' –> On my screen (at a view size of 100%) 1 m is equivalent to 6 mm, so the scale 1:100 might be correct in the original figure but it was changed when implanting in this document;

Reply: The scale below the word 'column' in the figure is deleted because the scale changes when zoomed.

Lines 148 and 150 (Fig. 2a and 2b): 'Total depth' means what? Depth of the borehole, 6 m?

Reply: Yes, 'Total depth' means 'depth of the borehole'. In practice, the borehole depths in the NSP and SRP are 15 m and 50 m, respectively. However, the depth of 6 m is enough for this paper because the maximum penetration of the active layer is less than 4 m. For the purpose of avoiding misunderstanding, the borehole depths are both changed to 6 m in Fig. 2.

Line 151: Mark figure parts either with a and b and refer to them in the caption or with NSS and SRS but not both;

Reply: The both figures are marked with a and b. They are referred to in the caption of Fig. 2.

Line 153 : : :: You are describing the equipment here but the heading of 2.1 only refers to the monitoring site;

Here, the constructing process of snow site is described, including the climate, vegetation, geology. As a part of this site, installation process is also introduce here. Description of the equipment is provided in the section 2.2 and 2.3.

Line 153: September 2009?

Yes. Before the snow site is established, it is used to study the characteristic of permafrost.

Line 154: '... meteorological measurement equipment ...'

Deleted.

Line 154: November 2009?

Revised.

Line 155: '... monitored using the sensors ...';

Revised.

Line 157: ': : : in May 2010 : : :'

Revised.

Line 157: Here you just refer to NSS and SRS. Explain your concept of investigating the effect of the snow cover by comparing one site with a natural snow cover with another one where you artificially removed the snow; this is essential for the understanding of the whole study; and describe it before you come to the measurement equipment;

Separation distance between the NSP and the SRP is approximately 300 m and thus ground surface and geological conditions can be regard as the same except snow cover. The

hydrothermal difference in the active layer between the NSP and SRP could be regarded as the result from seasonal snow cover.

Line 160: Ground temperature at which depth?

Please see the line 102.

Line 160-161: Do you mean: maximum duration of the snow season?

It was deleted here.

Line 160-161: Do you mean: maximum duration of the snow season?

It was deleted here.

Line 162: In line 153 you already mentioned a ground temperature equipment; how is it related to the probe you mention here?

Introduction of equipment is arranged in the setion 2.3.

Line 162: ': : : in May 2010.';

It is accepted in the line 132.

Lines 162-163: 'A set of ... was installed ... of the two plots, NSS and SRS, respectively

It is revised in the line 132 to 135.

Line 167: Why did you measure the soil thermal flux only at the NSS, but not at the SRS?

It may be a mistake. There are soil thermal flux probes at the both plots.

Lines 165-168: '... were added at the NSS and the SRS, while an ultrasonic snow depth sensor and a sensor measuring the soil thermal flux at shallow depths were

installed only at the NSS.';

It is revised in the line 132 to 135.

Line 170 (Table 1): Use small letter at the beginning; also for soil temperature;

Reply: Accepted.

Line 170 (Table 1): Under the name 'types' you are mixing parameters (e.g. AT&H) and measuring intsruments (e.g. barometer or rain gauge) –> harmonize, please; use singular, e.g. 'Type of probe';

Reply: Accepted. Please see Table 1.

Line 170 (Table 1): Here you list the measurement parameters wind speed, rel. air humidity, air pressure, precipitation, albedo in the methods but you don't present them in the results or discussion; If you don't need them for your paper delete them in Tab. 1;

Reply: Yes. Deleted measurement parameters are including air pressure, and albedo. Wind speed and direction and precipitation is mentioned in the discussion, RH is mentioned in the monitoring results.

Line 170 (Table 1): for the soil moisture probes you have to say if % refers to the gravimetric water content (as in Fig. 2) or the volumetric one;

Reply: Only in Figure 2 or content directly related with this figure, water content refer to gravimetric water content. In the other parts of this paper, water content and soil moisture both refer to volumetric water content. Description of water content has been noted in the caption of Figure 2.

Line 170 (Table 1): Why did you use different soil moisture and albedo probes at SRS and NSS? This makes it more difficult to compare the measured values;

Reply: Yes. It will be easier if the same sensors is used. But SRP and NSP are supported by different research projects. What's more, donors of the two projects are different.

Line 170 (Table 1): Please tell the measuring interval(s) of the different parameters;

Reply: They have been added. See Table 1.

Line 170 (Table 1): To me the following order of columns from left to right would be understandable more intuitively: Type of probe, Number of probes (NSS [as reference first], SRS), Measuring range, Measuring accuracy, Brand, Model;

Reply: Accepted. See Table 1.

Line 170 (Table 1): As not all parameters were monitored for the same period of time it would be helpful to give the monitoring period for each parameter, either in this table or even more vivid in a figure;

Reply: Monitoring period for each parameter is added in Table 1.

Line 170 (Table 1): 'Measuring range' instead of 'Ranges';

Reply: Accepted. See Table 1.

Line 170 (Table 1): 'Number of probes' instead of 'Numbers';

Reply: Accepted. See Table 1.

Line 170 (Table 1): In order to save space you could use small superscript numbers directly after the probe type;

Reply: Accepted. See Table 1.

Line 175: Not clear. Do you mean you used the measured air temperature to correct the signal of the ultrasonic sensor measuring snow depth?

Yes. But the context is deleted.

Line 177: Add space: 6 cm;

It is deleted.

Line 178: '... are thus needed ...'; then you can omit: 'because ... rapidly';

It is revise in the line 146-147.

Line 180: All means all, so you don't need to list some probes;

Accepted. It is revised in the line 136-139.

Line 182: Specify the logger by the brand, at least;

They are listed in the Table 1.

Line 182: '... to the automatic data logger CR3000.';

It was revised in the line 138-139.

Line 183: instead of 'that ... when' use 'after';

It is revised in the line 148.

Line 184: '... station, data were often not recorded during night at the SRS.';

It is revised in the line 149-150.

Line 185: '... area differing between seasons.';

It is deleted.

Line 186: But I can see a fence around the SRS in Fig. 3b and c; so what do you mean by snow fence?

Yes. There are fences at the both plots, which are not snow fence and only used to keep the animal away from the plot.

Line 188: How could you guarantee that the SRS wasn't covered by snow again due to wind after snow removal?

The snow cover is removed manually again if there is snowdrift.

Line 191: Distance between the three pictures are slightly not the same as well as their height –> adjust;

Reply: They have been adjusted. See Figure 3.

Line 193 : : :: A lot of information on the acquisition of soil temperature data was already given under 2.1; the information given here would fit to table 1; temperature measurement is a standard technique; why do you describe it here so extensively?

This part is revised in the section 2.4.

Line 193: 'soil temperature data acquisition' or 'soil temperature measurement';

Accepted in the section 2.4.

Line 205: See comments on the heading of 2.2 (Line 193) and transfer to 2.3;

See section 2.4.

Line 207: '... two parallel steel rods which are 300 mm long, 3.2 mm in diameter and separated by a distance of 32 mm.';

It is deleted.

Line 212-213: Did you dig the test pit and push the rods of the probes horizontally into a side wall or did you burry the probes in layers?

It is buried in layers due to the gravelly soil. This context is deleted.

Line 213: Not 'laid by drilling' but 'installed in a bore hole';

It was deleted.

Line 214: 'installed' instead of 'laid';

Accepted. See line 130.

Line 214: '... reaches its maximum thawing penetration.';

It is deleted.

Line 216: 'or' instead of 'and';

It is deleted.

Line 217: '... were thus due ...';

It is deleted.

Line 219: '... extended from the ground surface to 3.6 m depth?';

It is deleted.

Line 220: Not 'laid' but 'installed' (2 times in this line);

Accepted in the full text.

Line 222: Repetition, information already in Tab. 1;

They are deleted.

Line 223: What do you mean by hydrothermal probes? The soil temperature and moisture probes together? Did you have to dig into the active layer to install the mast onto which the ultrasonic snow depth sensors is mounted or why do you mention digging in connection with the snow depth measurements?

The soil temperature and moisture probes are separated. It was noted that the digging was carried out only in 2010, the influence of which on the active layer was very small. Installation of the mast did not involve digging in 2012.

Line 224: Not 'by' but 'in'

Accepted in the line 2.3.

Line 224: Instead of 'wouldn't' I would say 'shouldn't';

The part is deleted.

Line 226: '... water content in the thawed soil ...'?

Revised in the line 162-164.

Line 228: Do you mean because of the calibration for unfrozen conditions by the manufacturer? How did you get the true value? By gravimetric water content measurements using soil probes? Or do you stress the fact that the probes measure the liquid water content but not the frozen water content? Not clear;

Yes. The soil moisture probes only measure the liquid water content but not the ice. In order to discuss the true water content and its annual variability, measurement results in the thawing period are analyzed in the process analysis. The interannual analysis only involves SM measured in October 1 with the maximum thawing penetration, which represents the water content of the whole active layer and can be called the characteristic soil moisture.

Line 237: Do you mean accumulated in total?

Yes. See the section 3.2.

Line 237: Why do you say 'surface snow cover' instead of simply 'snow cover'? As the snow cover is always deposited on the surface this word is dispensable;

It is deleted in the full text.

Line 239 (Firg. 4): Curves are too thick (only in pdf?); add horizontal lines as visual orientation;

Reply: The figure is deleted.

Line 245: I question if the snow depth can be determined with a resolution of 1 mm (surface hoar on the snow can be thicker than 1 mm);

The accuracy of SCT probe is 1 cm. But its resolution is 1 mm. What's more, artificial measurement is also carried out before every snow removal.

Lines 246, 248, 252: Use past tense because it occurred in 2014; check whole manuscript and use past tense for all events which were finished in the past;

Accepted. See section 3.2.

Line 246: Check with editor if this date format is ok; However, the date format should be the same throughout the document which is not the case, e.g. line 245 (December 2013) differs from line 240 (2012.12);

Accepted and revised in the full text.

Lines 251-252: '... snow cover remained 5 days at maximum in more than 90 % of the snowfall events, ...';

Revised in the section 3.2.

Line 253: In the same sentence you say that the snow cover would have been lasting not more than 5 days in general - this is a contradiction; instead of the words 'generally' and 'typically' try to quantify in the statements;

Revised in the section 3.2.

Line 260: Add space after fullstop;

Revised in the section 3.2.

Line 270 (Tab. 2): Data of air temperature and radiation averaged over the seasons would be helpful here as you argue in the text with these parameters;

Reply: Accepted. Averaged air temperatures and average solar radiation over the seasons are added. See Table 2 and table 4. See section 3.1, 3.3.

Line 272: As you define the depth negative in Fig. 7 the maximum depth would be 0; so either you can say something like 'the most negative depth' or you use a positive sign for depth values in Fig. 7;

A positive sign for depth values is accepted in Fig. 7.

Line: 277: 'Figure 7' but not 'figure 7';

Accepted. It is revised.

Line 277: I question if the ALT can be determined with a resolution of 1 mm;

The ALT is calculated based on the interpolation between temperature measurements of neighboring probes in space and time. Resolution of the ALT depends on the space of probes and frequencies of measurement. Below the depth of 3.0 m, separation distance of neighboring temperature probes is 1 m. Due to the difference of soil properties and unsteady temperature regime, distribution of temperature with the depth is often nonlinear. Therefore, accuration of the ALT should be far more than 1 mm.

Additionally, the ALTs in the discussion manuscript are acquired by using 10 days step in the interpolation calculation. In this revised manuscript, time step is 1 day. As a result, there are some differences between the ALTs of this version and previous version. The maximum difference reaches 14 cm.

Line 280 (Fig. 7): Indicate at least one negative isotherm;

Reply: Accepted. See Figure 7.

Line 280 (Fig. 7): Mark figure parts either with a and b and refer to them in the caption or with NSS and SRS but not both;

Reply: Accepted. See Figure 7. a) natural snow plot, b) snow removal plot.

Line 286: Regarding 'at any depth' –> the mean annual ground (rather than soil) temperature MAGT refers to the depth of the zero annual amplitude (see Wu and Zhang, 2008, caption of Table 2);

Yes. It was changed to " the mean annual soil temperature (MAST)".

Lines 284-288: This sentence needs to be related to the ground temperature measurements of this study;

It was introduced in the section 2.5.

Line 290: '... where the daily soil ...';

It was introduced in the section 2.5.

Line 291: Do you mean the extremes of daily air temperatures within a year? This must become more clear because the extremes of the air temperature within a day must not be excluded to estimate the daily geothermal propagation depth;

It was introduced in the section 2.5.

Line 293: Did you determine the daily geothermal propagation depth? If not, please cite the publication where you got this information from;

It was introduced in the section 2.5.

Line 294: It depends on the purpose; what do you want to say with these data?

It was introduced in the section 2.5.

Lines 293-296: I understand the sentence up to '... single time', but the words thereafter (several times, even partial time of each day) I can't logically relate to the rest of the sentence. Maybe 'or' is missing before 'even'?

It was introduced in the section 2.5.

Line 298: 'could' instead of 'can';

It was revised in the section 3.5.

Line 301: 'also' is dispensable;

It was revised in the section 3.5.

Line 303 (Fig. 8f): How do you interprete that the MAGT-curves are bent towards lower temperatures above 1.5 m? Compare to Fig. 3 in Smith & Riseborough (2002): Climate and the limits of permafrost;

Reply: Generally, MAGT-curves should be similar with that in Fig.3 in smith & Riseborough (2002): Climate and the limits of permafrost if the ground thermal is stable and water content is high in the active layer. However, owing to the variation of MAAT, earth surface conditions and soil moisture in the active layer, the MAGT-curves often behave different shapes. Taking an example of permafrost along the Qinghai-Tibet highway, there are three typical shapes of MAGT-curves (Jin et al., 2006. Thermal regimes and degradation modes of permafrost along the Qinghai-Tibet highway. Science in China Series D: Earth Sciences. DOI: 10.1007/s11430-006-2003-z ). The mean annual air temperature, water content in the active layer and the snow cover changed a lot in the snow monitoring site since 2011. The MAAT fluctuated between - 4.5 ℃ and –2.8 ℃ during the period of 2010-2016. Snow cover is removed since the December, 2012. Changes of surface conditions caused that the 5cm-MAGT in the SRP is about 0.2 to 0.8 ℃ lower than that in the NSP. Additionally, volumetric water content above the depth of 0.8 m has been nearly always lower than 12% in the SRP from 2012 to 2015 (Figure 10). So, thermal offset in this soil layer can be neglected. Drop of MAAT and 5 cm-MAGT, and the shallow soil layer with less water content, may be the main reason that the MAGT-curves are bent towards lower temperatures.

Line 303 (Fig. 8f): I recommend not to choose blue and red lines in Fig. 8d as these colors are used in a different meaning in Fig. 8a - 8e;

Reply: Accepted. See Fig.8f.

Line 303 (Fig. 8): The colors in Fig. 8 should be harmonized with those in Fig. 10 regarding their meaning;

Reply: The colors are modified according to the year. Blue, 2012; red, 2013; black, 2014; purple, 2015; orange, 2016. See the detail in Fig.8 and Fig.10.

Lines 304-307: In line 304 you say 'annual' for the entire Fig. 8, but later you say that the temperature profiles are averaged over seasons!

Reply: It has been revised. See the caption of Fig.8 and 9.

Lines 309-311: This should additionally be described in methods;

Reply: It was revised in the section 2.5.

Line 323: 'In terms of yearly temperature,' is dispensable;

It was revised in the section 3.5.

Lines 323-325: As first sentence of this paragraph I suggest: 'The mean annual soil temperature at 0.5 m depth was 0.8 _C higher for the NSS than the SRS while the temperature difference between the two sites decreased with depth being approximately zero below 2 m depth.';

It was revised in the section 3.5.

Lines 325-326: According to Figure 8e they are approximately the same compared to the layer above 1.6 m!

It was revised in the section 3.5.

Line 327: What do you mean by 'generally' here? Average difference over the whole depth profile or only below 1.6 m? Not clear;

It was revised in the section 3.5.

Lines 328-329: This sentence in the present form belongs to methods but not to results; don't

repeat methods in results;

It was revised in the section 3.5.

Lines 329-330: '... removal, the mean annual ground temperature at 0.5 m and 2.0 m depth increased by 0.3 _C and 0.2 _C, respectively.';

It was revised in the section 3.5.

Lines 323-333: Not clear! I suggest: 'The MAAT in Yashatu was -4.5 _C, -3.4 _C, and -3.9 _C in 2012, 2013 and 2014, respectively, indicating ...';

It was revised in the section 3.5.

Line 333: Where did you get the value for 2012 from? According to Fig. 4 you only measured the air temperature since December 2012; you have to cite the source if it is not your own measurement;

It was revised in the section 3.5.

Line 334: No comma after 'namely';

It was revised in the section 3.5.

Lines 338-339: This sentence is only valid for NSS, tell the reader;

It was revised in the section 3.6.

Lines 340-341: Regarding '..., there is ...' –> Where? Do you mean at the SRS? So you have to tell it the reader. However, I see something different in Fig. 9: besides the small L-shaped area at SRS with a water content above 40 % in 2013 the water contents at NSS are higher throughout the depth profile and throughout the monitoring period;

It was revised in the section 3.6.

Line 342: 'based on CS616' does not belong to results but to methods and shouldn't be repeated;
It was introduced in the section 2.4.

Line 342: It can, but does it or does it not?

It was introduced in the section 2.4.

Lines 343-345: There appeared strong changes during redistribution, but if you directly compare the water content profile at NSS in Oct. 2013 and Oct. 2014 I roughly agree although I can see an increase from 30 to 40% between about 1.1 m and 1.7 m depth –> has to be described more clearly;

It was revised in the section 3.6.

Line 345: Is the class with the highest water content from 40 to 70%? Please indicate in the color bar in Fig. 9. However, a vol. water content of 70% in gravel (as stated in Fig. 2) is very unlikely to me even for saturation (maybe in clay it would be possible); Line 349: Just to describe what you can see anyway in Fig. 10 doesn't give an additional benefit; add information in the text you can't directly see in a Fig. itself;

It was revised in the section 3.6.

Line 349: Don't make a new paragraph after only 1 sentence (minimum after 2);

It was revised in the section 3.6.

Lines 350-351: 'At a depth of 0-50 cm soil moisture sites varies no more than 4 % between the years 2012-2014 at each of the two sites.';

It was revised in the section 3.6.

Line 351: 'soil moisture decreases with time, and' –> not needed, delete;

It was accepted in the section 3.6.

Line 354: 'Between 80 cm and 120 cm depth, soil ...';

It was revised in the section 3.6.

Lines 354-355: Sentence is too complicated to express the facts;

It was revised in the section 3.6.

Line 356: Give a depth range as in Fig. 10;

It was revised in the section 3.6.

Line 358: 'first' and 'then' means from 2012-2013 and from 2013-2014, respectively; express it more clearly;

It was revised in the section 3.6.

Line 360 (Fig. 9, caption): In Table 1 you say that soil moisture at NSS was measured using a probe called SM300 but not CS616;

Reply: It has been revised. See the caption of Fig.10.

Line 360 (Fig. 9, caption): ... 'based on CS616' is an information regarding the methods and must not be repeated here;

Reply: It has been revised. See the caption of Fig. 10.

Line 359 (Fig. 9): How do you explain the pronounced steps in the water content profiles, especially in 2013?

Reply: In the Fig. 10, 0 ℃ isotherm is added. It can be seen from the figure that there are often big steps in the water content near the 0 ℃ isotherm. It can be explained that the dielectric constant of water is much less than that of ice. In the thawed area, water-content step also can be found. It be resulted from the downward infiltration and upward migration of soil water.

Line 359 (Fig. 9): What is the upper limit of the highest class above 40 %? Please indicate on the color bar.

Reply: The maximum of water content of the active layer in the snow site is 59.7%. It was added on the color bar.

Lines 364-369: Too complicate; just say that you linearly interpolate between the point measurements using the soil moisture probes;

A good suggestion. Accepted. See line 324 to 328.

Line 372: '... at 0-5 cm depth is assumed to be the same as ...';

Accepted in line 330.

Lines 364-379: This information would be more adequate in the methods;

I think this is a reprocessing of field data. It may be coherent with the context. See line 326 to 333.

Line 379: A range is from a minimum to a maximum value; here you mean the difference;

It was replaced by new analysis in section 3.6. See line 334-343.

Line 382: Not 'in 2013 and 2014' but 'from 2012 to 2013 and from 2013 to 2014';

It was revised in the section 3.6.

Line 385: Mark figure parts either with a and b and refer to them in the caption or with

NSS and SRS but not both;

It was accepted in the Figure 11.

Line 390: Add space after Alps;

Accepted in line 348.

Line 391: Don't use first names for the citations in the text (only initials of first names in bibliography);

Accepted in line 349.

Line 394: See comment on line 47 and 51;

Accepted in line 351-352.

Line 397: Cite these studies;

The part is deleted.

Line 399: It did in case of ASR-1, but not for ASR-2;

The part is deleted.

Line 401: A comma or 'and' is missing after 'snow removal';

The part is deleted.

Line 402: To see the effect of the snow cover at the SRS by comparing the 3 years 2012, 2013 and 2014 (i.e. BSR, ASR-1, ASR-2) is hardly possible as you have 3 samples (years) only, but 2 factors (air temperature and snow cover);

The part is deleted.

Lines 400-403: You cannot directly compare ASR-2 (i.e. SRS 2 years after snow removal) with NSS because the observation period of ASR-2 was 2013.12-2014.11 (line 311) while it was 2014.3.1-2015.2.28 (line 300) for NSS; however, you can compare SRS and NSS in the same period of monitoring as shown in Fig. 8e. Maybe you meant this, but it is not clear;

The part is deleted.

Line 405: '... heat dissipation from the active layer to the atmosphere in winter (...).';

Accepted in line 360.

Line 407: Omit 'the';

The part is deleted.

Line 407: Are these results yours? If not, cite the publication in the same sentence;

The part is deleted.

Line 409: This finding means that the ground cooling effect due to the change of the albedo by the first snow cover in autumn is less effective than the cooling due to the stronger heat dissipation without snow; finally, the thermal insulation (and during this season warming) effect of the snow cover overbalanced the cooling effect due to the higher albedo of snow in this study;

Yes. Monitoring result in 2015 and 2016 also verified this findings.

Line 411: In addition to Fig. 6, also refer to Table 2;

Accepted.

Lines 414-415: '... when the ground temperature is higher than the air temperature'?

Accepted in line 371 and 372.

Line 418 (Table 3, caption): The table is obviously not at the right place but should be positioned later;

It is deleted.

Line 418 (Table 3, caption): This is again another period of time than for the collection of the other data (soil temperature, air temperature, water content); so it is difficult to relate them to each other;

The part is replaced by new analysis. Please see line 380-410.

Line 418: Table 3 shows results and should thus be presented in chapter 3, but not only in the discussion;

It is deleted.

Line 421: '... since the snow ...';

The part is deleted.

Line 423: '... of the active layer.';

The part is replaced by new analysis. Please see line 384-395.

Line 424: Which areas?

The part is replaced by new analysis. Please see line 384-395.

Line 426: not correct: latent heat is released by freezing and by condensation, but not by a decrease in the water content; if the soil water content in autumn is less than the year before, also the latent heat released during freezing will be less; is it this point you wanted to make? Then make it clearer, please

The part is replaced by new analysis. Please see line 384-395.

Line 427: See comment on line 50;

Corrected.

Line 429: But at 4000 m a.s.l. altitude the pressure is much lower –> adapt the following calculations to a realistic atmospheric pressure at 4000 m a.s.l.;

The part is replaced by new analysis. Please see line 384-395.

Line 429: '... heat for water stored in a 1 m3 soil body at a volumetric water content (VWC) of 1 % is 3350 kJ ...';

The part is deleted.

Line 431: How did you determine this heat capacity? Please show that it is realistic using the content of mineral material, ice and water;

See line 389-390.

Line 430-433: Has to be explained more clearly. Just by reducing the water content the temperature won't decrease! However, the heat to be extracted from 1 m3 of soil to freeze the water equivalent to a VWC = 1% is 3350 kJ. If the same heat would be extracted from the same body of soil without freezing (i.e. if already al the water is frozen or al the water remains liquid) a temperature decrease of 1.5 _C would occur. For the heat of vaporization the argumentation is equivalent;

Please see line 384-395.

Line 435 : : :: I can't see that you used the thermal flux data to verify 'this phenomenon' explained above. I would just argue that at higher water contents more heat has to be extracted for freezing than in drier soils. The atmosphere can take up only a given amount of heat under given meteorological conditions. After freezing, in driers soils more heat that can be dissipated is left for ground cooling resulting in lower temperatures;

The part is deleted.

Line 437: Add a space before 'According';

Corrected.

Line 438: To make these four stages more visible I suggest a diagram rather than a table;

The part is deleted.

Line 444: It should be like this, but Table 3 shows the opposite! You argued for all other months. I would say the higher/lower values in February/June at the NSS than at SRS is just natural variability;

The part is deleted.

Line 446: Even greater than in Table 3? And why should the heat exchange be different at NSS and SRS if there is no snow? Because of the higher water content at NSS than at SRS? Then

you need to argue which processes lead to the assumed result; however, it has to be consistent with your other findings;

The part is deleted.

Line 456: Instead of the grain size distribution it is rather the pore size distribution;

Accepted. See line 428.

Line 457: Why and how? I would say the pore size distribution, the porosity and thus the bulk density are changed by digging, but not the grain size distribution;

I agree with you. See line 427-430.

Line 462: Is it possible that the digging lead to preferential flow paths in the ground? However, you dug at both sites in the same way, right?

I agree with you.

Line 464: 1 space before 'therefore' is engough;

Accepted.

Line 469: '...), resulting in a snow water equivalent (SWE) of ...';

It is revised. See line 440-441.

Lines 471-473: Does the annual rainfall of Delingha originate from 1960? Or was it only published then and is a long term mean, i.e. even older? It could have changed a lot since then! So you can't compare it with todays values of Yashatu;

It is revised. See line 415-416.

Lines 480-481: Is half a page really necessary to come to this explanation?

It is revised. See section 4.2.

Lines 483: Repetition of line 469 –> delete;

It is revised. See section 4.2.

Line 483: 'the melt water equivalent to the SWE' instead of 'this result';

It is revised. See section 4.2.

Line 484: '... content between 0-2.5 m depth in the active layer ...';

It is revised. See section 4.2.

Line 483: '... could have increased ... by only 4.4 % ...';

It is revised. See section 4.2.

Line 493: Only in summer, when the evaporation is highest, the NSS was cooler than the SRS at 0.5 m depth (Fig. 8). However, then there was nearly no snow (Table 2). So I don't think that this effect contributed to the lower water content at SRS, otherwise the near surface temperatures (0.5 m depth) should have been higher at SRS than at NSS also in spring, winter and autumn which was not the case;

Before 2014, the temperature of topsoil above the depth of 0.5m was not acquired due the acquisition frequency of one time per day. According to the data in 2015 and 2016, the seasonal and annual topsoil temperature at the SRP is lower than at the NSP except for the autumn (Figure 8 and 9).

Line 497: But according to Tab. 2 there was nearly no snow in summer at your site;

It is revised.

Lines 498-499: But in Fig. 8 you have shown that the mean winter temperature is lower at the snow free SRS than at the snow covered NSS!

It is revised. See section 4.2.

Line 504: Also this sounds logical but should have resulted in lower near surface temperatures

at NSS than at SRS, also in winter, spring and autumn. Or do you think this was the case, but you couldn't show it as no data could be gained above 0.5 m depth? Then you have to say it

It is a wrong conclusion. New analysis is displayed in the 4.2. See line 453-463.

Line 507: Which range do you mean - within a year? Then during the first year the VWC ranged between 0 and the highest class (40 - ?%), in the second year between 0 and the second highest class (30% - 40%). So what do the 50 % decrease mean?

The part is replaced by new analysis. Please see line 453-463.

Line 508: Add a space after the fullstop;

Corrected.

Line 515: The mean surface?

It is revised. New conclusions were drawn according the field data in 2015 and 2016.

Lines 523-525: The topic of your paper are the effects of the seasonal snow cover on the hydrothermal conditions of the active layer; so here you should conclude that the snow removal at SRS lead to lower water contents which can be derived from the comparison with the NSS where the measurements took place at the same period of time facing the same (dry) meteorological conditions. So far (in point 3 of the conclusions) you argue only by the temporal sequence of 3 years;

I agree with you. New conclusions were drawn according the field data in 2015 and 2016.

Line 524: '... SRS can be attributed to the removal of ...';

It was replaced by other conclusions.

Line 628: Check that the authors' family names and the initials are always separated by commas in the bibliography (which is not the case e.g. in line 628 and 630)

It was replaced in line 631.

---

## Author Comment (AC2) · 15 Jun 2017

**Reply to RC2**

**Replies to the general comments:**

1. The introduction chapter includes a very detailed literature review. I recommend to shorten this part, and only include the background necessary to put the paper in a larger context. Highlight why this study is unique and needed in context of previous studies on the same topic in the introduction, but avoid starting the discussion here. Rather move parts of it (with many of the references) to the discussion chapter where you discuss the results in relation to previous findings.

The introduction is sharpened to one page. Some parts are moved the discussion section.

2. I miss a presentation of the objectives in the paper. Please include clear objectives, e.g. in the last paragraph of the introduction chapter.

The objectives are re-organized and presented as a single paragraph at the end of the introduction.

3. The different observation periods for each variable is confusing for the reader. Please make a figure or table illustrating the period of measurement for each vari able/instrument, e.g. as a timeline of observations. In addition, I cannot find explicit information on the period when the snow removal is done. I assume this was done for the entire period 2012 to 2015? This is crucial information in this paper! Please specify.

Monitoring period of all parameters is pictured and uniformed. Please see the Figure 4.

4. The authors state that the temperatures at NSS are warmer than at SRS over a calendar year, and suggest that removal of the snow has a cooling effect on the ground. However, I miss a clear quantification of the difference between the sites and how this changes with time, supporting these statements. Does the difference increase by each year? Functions of running mean annual temperatures of some selected temperature loggers (depths) would be useful, as well as MAGT for each year at each site.

The soil temperature in the active layer at the NSP (NSS) and SRP (SRS) were reanalyzed according to the new monitoring data in 2015 and 2016. The current conclusions are very different with the previous. Please see the section 3.5.

5. How can the ALT be determined with an accuracy of mm in the range 3.4 - 3.6 m, when the soil temperature measurements are only located at 3 m and 4 m depth? These depths are derived from (I assume linear?) interpolation of the temperature logger data. Because of variation in ice content and ground material this may not be entirely true, and the use of mm precision does not make sense. The ALT derived from interpolation can therefore not be used to differentiate the change in ALT between the two sites. I would say it could perhaps give an indication of ALT thickness within 10 cm, but it has to be noted in the paper that this is an approximation. By this you could still say that ALT at both sites are increasing, but you cannot differentiate the ALT change. In order to assess the differences between the sites, please compare observed temperatures at 3 and 4 m depths between the sites.

The ALT is calculated based on the linear interpolation between temperature measurements of neighboring probes in space and time. Resolution of the ALT depends on the space of probes and frequencies of measurement. Below the depth of 3.0 m, separation distance of neighboring temperature probes is 1 m. Due to the difference of soil properties and unsteady temperature

regime, distribution of temperature with the depth is often nonlinear. Therefore, accuration of the ALT should be far more than 1 mm.

Additionally, the ALTs in the discussion manuscript are acquired by using 10 days step in the interpolation calculation. In this revised manuscript, time step is 1 day. As a result, there are some differences between the ALTs of this version and previous version. The maximum difference reaches 14 cm.

The early two year data is not enough to differentiate the difference of ALT change between the both plots due to the small change. However, the ALT was stable since 2015 at the NSP, but the ALT increased by more than 100 cm since 2014.

6. The actual effect of the snow removal on ground temperatures is not clear to the reader (see points 5 and 6). It is therefore also difficult to follow the discussion of why snow removal has a cooling effect. However, IF the effect is cooling at SRS compared to NSS, the discussion must focus on establishing the cause of this effect. Is the reason a change in thermal insulation, albedo, efficiency of longwave radiation exchange, energy lost to snow melt or infiltration of meltwater/soil moisture (see e.g. summary in Zhang, 2005)? In most areas with a developed snow cover the first effect (thermal insu lation) would dominate, and the result of snow removal would be cooling of the ground. However, as the authors correctly highlight, a 5 cm thick snow cover is normally considered too thin to have an insulating effect on the ground. Still, the authors spend most of the paper discussing whether the thermal insulation from snow is the reason for the cooling effect. As a reader, I would really doubt that this is the case, and therefore it is crucial that you support this discussion with observations. The most obvious would be to compare hourly temperature observations from the air to the uppermost logger (5 cm) in both boreholes. In this way you could see if there is a pronounced dampening of the daily temperature amplitude after a snowfall at 5 cm depth at the NSS site, and not at the 5cm observations at the SRS site.

According to your suggestions, thermal effect of snow were analyzed again by adopting new high frequency data, including ground surface temperature (by Apogee), topsoil temperature. The new conclusions are satisfied. Please see the section 3.3, 3.4, 3.5, and 4.1.

7. The entire discussion and logical structure behind arguments has to be improved throughout the paper. It is difficult for the reader to relate the discussion around effects to the presented results. The soil moisture data presented in Figure 10 shows interesting results, with a general drying of the SRS site. There is rapid decreases at some of the depths, which has to be commented upon. What is the accuracy of these observations? The authors also link the drying of the ground to reduction of melt water infiltration and increase of evaporation at the SRS site. This seems very likely, but I miss the explicit link from this to the thermal effect it would have on the ground. In general, a soil with less moisture would reduce the exchange of latent heat, as latent heat from freezing of soil moisture is a large energy source, while melting of ice is a similarly large energy sink. Over a year the latent heat energy in and out of the system would be equal, if the soil moisture is not changed. However, in this case there is a gradual drying of the ground at the SRS site during the period, potentially resulting in more melting and evaporation than freezing of water during one year. This would be an energy sink and consequently cool the ground. The authors comment both of these effects, which seems very likely in light of the observed soil moisture data. Still, the authors spend more time discussing the effect of thermal insulation in the discussion chapter. This seems strange, as the presented

data indicate that the reduced latent heat effect from drying of the soil may be an explanatory factor, and there are no results demonstrating that the thermal insulation from the snow cover play a major role.

According to this suggestions, the cause of the rapid decrease of soil water were discussed in detail (line 447-464, 476-486). The thermal effect of soil moisture loss on the active layer were also analyzed fully. Please see the line 385-411.

8. In Figure 4 you present the relative humidity observations. However, these are used only to give the average annual relative humidity. It would be nice to connect these observations to the discussion of the effect of increased evaporation in chapter 4.2.

From 2012 to 2016, the RH increased by approximately 7%. This change of RH can explain the decrease of soil moisture in the first 2 years. However, it should act as an assistant role because its effect on the both plot is the same. In fact, the CSM reduction at the SRP is higher 8.2% than at the NSP. So, its effect was not given here.

9. The paper would benefit from a better structure and consistency. Some results are mixed into methods chapter (line 138- 146 and line 159-161). The logical structure of the discussion points also has to be improved. Please refer to the results when you discuss them, and ideally quantify the finding, i.e. not only refer to "a cooling effect", but give how much cooling compared to reference site.

Structure, content, and discussion method all changed a lot comparing with the last version. More importantly, some new findings were got during this process.

10. Please keep a consistent time format throughout the paper. At least four different date-formats are used in figures and in the text. Please clear this up!

The time format is uniformed.

11. Figure 4 to 6 and also ideally Figure 7 should be presented together (in the same figure or below each other at one page) with the same date format so that the data can be related to each other! Now they represent different periods of observation, all have different date format, and they are not in the same figure.

The periods of observation are the same in the Figure 5, 6, and 7. But I did not find an ideal way to put them in one picture because they exhibit the different content. Perhaps, the copy editor could give a good suggestion.

12. The text would benefit from a simpler and clearer language. Some sentences are lengthy and could be simplified. This may partly be a language problem, but the general content could also be sharpened.

We make some attempts in this version. It is expected that it can do.

**Replies to the specific comments:**

Line 21: "Maybe" in the abstract is a bit vague. Rather use "likely", or "we believe". This is also valid for the conclusions (point 3, L 524).

It was corrected.

Line 21: What do you mean with "the delay of snowfall time in autumn"? Please clarify, and relate it to the physical process – does this also refer to the insulating effect of snow, or other effects?

It was deleted in this version.

Line 53: Change into "Low thermal conductivity of snow". Delete coefficient. What do you mean with "The Alps"? The Swiss Alps?

The part was deleted in this version.

Line 58 – 85: Why is the observed effect, given in degrees C, of variation in snow depths higher in some areas than in others? I would expect this to be directly coupled to the climate (very cold winters or maritime mild winters?) Since snow cover with a critical thickness (typically 60-80 cm depending on the snow properties) disconnects the ground surface temperatures from the air temperature due to very low thermal conductivity, it follows that the difference in ground temperatures between a site with 40 cm snow and a site with 80 cm thick snow is closely related to the air temperature. A very cold winter would result in a large difference, while a warmer (maritime) winter with temperatures close to 0C would result in less difference.

The part was shortened greatly in this version, the objective of which is to learn the thermal effect of snow cover with different thickness. Study on the thin snow cover was still scarce.

Line 56: Thermal conductivity is normally given as W/m K, or better W m-1 K-1, where Kelvin is denoted with capital K. The latter notation is used later in the paper; please be consistent. Also clarify the meaning of "d" in W/m K d. If this is temporal rate change of thermal conductivity per day, change into W/m K day or W m-1 K-1 day-1.

It is temporal rate change of thermal conductivity per day. The part "W/m K d" and relative content was deleted in this version.

Line 77: Here I would also refer to Haeberli and the "Bottom temperature of snow" (BTS)-method.

This method is valid in the thick snow cover. It may be not appropriate on the QTP due the thin and short-term snow cover.

Line 89 and 90: What is the permafrost "shell"? Please clarify.

It refer to the perennial frozen soil layer. This part was deleted.

Line 92-94: I guess there are also several newer models developed for this purpose. Is there a reason why you mention this old one in particular? If not, please remove.

It was removed.

Line 98-99: I don't understand the meaning here. Do you mean "a wide distribution of snow depths"?

It should be "a wide distribution of snow cover", the aim of which is to tell the readers that there are snow here.

Line 100: snow covered days

This part was deleted.

Line 100-101: gradual increase in the height of the stable snow cover

I thought that the snow cover area increased. Now, it was deleted.

Figure 1b: This figure does not tell much. Either leave out and give distance between the sites and the elevation in the text, or include some background information on vegetation type or similar.

It was replaced by a google map, which can include some information on vegetation type.

Line 131: suggested change: "average annual temperatures"

The part was given in the section 3.1.

Line 142: propagation depth at the snow site

This part was deleted.

Line 138 – 161: The description of the monitoring site (2.1) is quite lengthy. This is partly due to results mixed into the methods chapter (e.g. line 159 – 162). Avoid this and be more concise.

It was revised. The description on the ground temperature and snow was deleted.

Line 177-178: This sentence does not fit in the methods chapter.

This part was deleted.

Line 186-188: This is an important part of the method (the removal of snow) which is hidden away.

It was described individually in the section 2.2.

Line 201 – 208: This information is included in the table, and it is therefore not necessary to repeat it in the text.

This part was deleted.

Line 274: What is "thawing and freezing process curves"? Please be more precise, e.g.: "continuous plots of interpolated ground temperatures"?

Accepted in the line 255-256.

Line 271-278: It is enough to specify how you determine the active layer thickness and the actual thickness; e.g. "Continuous plot of interpolated ground temperatures for the period xx to xx are shown in Figure 7. Here we define the active layer thickness as the maximum depth of the 0 C isotherm (Muller, 1974). From the continuous plots we find that the ALTs of the two sites are xx cm and xx cm in 2013 and xx and xx in 2014."

Accepted in the line 252-254.

Line 291 – 296: This part is unclear. Either refer to observations or cite previous studies.

This part was rewrote and moved to the section 2.5 as method introduction.

Line 299: Suggested change: "Profiles of seasonal average soil temperatures interpolated between the loggers from 0.5 m to 4 m depths at SRS and NSS are shown in

Partially accepted in line 265-266, 281-282.

Figure 8. The averages are made over the period 2014-03-01 – 2015-02-28."

Partially accepted in line 269-271, 285-287.

Figure 8: Inclusion of the season (e.g. Mar – May) on each plot would make it easier to read. Similar x-axis would also make it easier to see the relative variation in temperature differences. In Figure 8f (or in a separate figure) it would be good to also include the evolution of temperatures in the NSS. This would make it easier for the reader to understand if the change in SRS is only due to climatic changes, or if it is due to the removal of snow, and if the initial situation at the two sites where similar or if the variation where as large as in figure 8e also in 2011-2012.

It was rewrote in line 285-287.

Line 350 – 358: Do you refer to Figure 9 or 10 here? Please specify and make references in the text. Also make sure you highlight interesting points from the figure, and don't reproduce the figure in the text.

This part was rewrote in line 301-315.

Line 371: Do you have a reference on Eq. 1?

I have no. It is only a formula based on the linear interpolation.

Figure 10: Specify that this is soil moisture at maximum thaw penetration (October) each year. Also consider placing this figure together with Figure 9, and indicate the timing of the

calculations in Figure 10 with lines in Figure 9.

Introduction on the data in this figure is given in the section 2.5. (line 188-191)

Line 396-399: Repetition form introduction. Please reduce the amount of redundancy.

This part was deleted.

Line 399: The ground temperature in the SRS should therefore increase after snow removal.

This part was rewrote in line 348-354.

Line 400 – 403: Repetition. Delete "the thickness of the snow cover was smaller than the critical snow cover thickness" and include "snow removal, while the average soil temperature"

Accepted.

Line 404-405: You state above that thermal insulation from the snowpack is not a dominating effect with snow heights lower than 20 cm. Here you still argument that this could possibly be an effect. Why do you believe so? You have to support this with observations! Again, this can e.g. be done by comparing daily temperature amplitudes in the air and in the topsoil (5 cm depth) before and after snowfall, at the SRS and at the NSS.

New explanation was given. Please see line 355-370.

Line 407-408: Please include a reference for this statement, or clarify if statement refers to the same study as referred above.

This part was deleted.

Line 408 – 417: Is this applicable if the snow cover is NOT thick enough to have an insulating effect?

This part was rewrote in lines 359-375.

Table 3: Is snow clearing still in effect in 2015 and 2016? Please be explicit.

The table is deleted.

Line 434: The "other significant factor" than what? Thermal insulation? There are far more indices that the decrease in soil moisture is an effect than the thermal insulation, which has no effect proven from the data.

This is an expression problem. We want to say that the dramatic decrease in soil moisture is the second reason besides the thermal insulator of the snow cover.

Line 435 – 449: Please relate the differences in fluxes to physical processes. Please be more explicit.

This part is deleted totally in this new version.